# $\psi$DAG: PROJECTED STOCHASTIC APPROXIMATION ITERATION FOR DAG STRUCTURE LEARNING

## ABSTRACT

Learning the structure of Directed Acyclic Graphs (DAGs) presents a significant challenge due to the vast combinatorial search space of possible graphs, which scales exponentially with the number of nodes. Recent advancements have redefined this problem as a continuous optimization task by incorporating differentiable acyclicity constraints. These methods commonly rely on algebraic characterizations of DAGs, such as matrix exponentials, to enable the use of gradient-based optimization techniques. Despite these innovations, existing methods often face optimization difficulties due to the highly non-convex nature of DAG constraints and the per-iteration computational complexity. In this work, we present a novel framework for learning DAGs, employing a Stochastic Approximation approach integrated with Stochastic Gradient Descent (SGD)-based optimization techniques. Our framework introduces new projection methods tailored to efficiently enforce DAG constraints, ensuring that the algorithm converges to a feasible local minimum. With its low iteration complexity, the proposed method is well-suited for handling large-scale problems with improved computational efficiency. We demonstrate the effectiveness and scalability of our framework through comprehensive experiments, which confirm its superior performance across various settings.

## 1 INTRODUCTION

Learning graphical structures from data using Directed Acyclic Graphs (DAGs) is a fundamental challenge in machine learning (Koller & Friedman, 2009; Peters et al., 2016; Arjovsky et al., 2019; Sauer & Geiger, 2021). This task has a wide range of practical applications across fields such as economics, genome research (Zhang et al., 2013; Stephens & Balding, 2009), social sciences (Morgan & Winship, 2015), biology (Sachs et al., 2005a), and causal inference (Pearl, 2009; Spirtes et al., 2000). Learning the graphical structure is essential because the resulting models can often be given causal interpretations or transformed into representations with causal significance, such as Markov equivalence classes. When graphical models cannot be interpreted causally (Pearl, 2009; Spirtes et al., 2000), they can still offer a flexible representation for decomposing the joint distribution.

Structure learning methods are typically categorized into two approaches: score-based algorithms searching for a DAG minimizing a particular loss function and constraint-based algorithms relying on conditional independence tests. Constraint-based methods, such as the PC algorithm (Spirtes & Glymour, 1991) and FCI (Spirtes et al., 1995; Colombo et al., 2012), use conditional independence tests to recover the Markov equivalence class under the assumption of faithfulness. Other approaches, like those described in Margaritis & Thrun (1999) and Tsamardinos et al. (2003), employ local Markov boundary search. On the other hand, score-based methods frame the problem as an optimization of a specific scoring function, with typical choices including BGe (Kuipers et al., 2014), BIC (Chickering & Heckerman, 1997), BDe(u) (Heckerman et al., 1995), and MDL (Bouckaert, 1993). Given the vast search space of potential graphs, many score-based methods employ local heuristics, such as Greedy Equivalence Search (GES) (Chickering, 2002), to efficiently navigate this complexity.

Recently, Zheng et al. (2018) introduced a smooth formulation for enforcing acyclicity, transforming the structure learning problem from its inherently discrete nature into a continuous, non-convex optimization task. This formulation allows for the use of gradient-based optimization techniques, enabling various extensions and adaptations to various domains, including nonlinear models (Yu et al., 2019; Ng et al., 2022b; Kalainathan et al., 2022), interventional datasets (Brouillard et al., 2020;

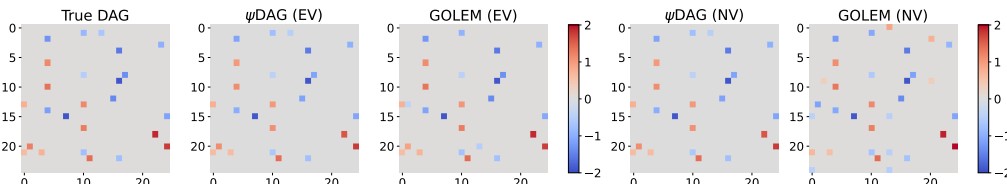

Figure 1: Visual comparison of the learned weighted adjacency matrices on a 25-node ER2 graph under Gaussian noise with equal variances (EV) and non-equal variances (NV, with noise ratio $r = 5$). For both methods $\psi$DAG and GOLEM the $L_1$ distance in the EV setting is 2.6. In the NV setting, $\psi$DAG maintains an $L_1$ distance of 2.6, while GOLEM's $L_1$ distance increases to 10.7, highlighting the robustness and generalization ability of $\psi$DAG across varying noise conditions.

Faria et al., 2022), unobserved confounders (Bhattacharya et al., 2021; Bellot & Van der Schaar, 2021), incomplete datasets (Gao et al., 2022a; Wang et al., 2020), time series analysis (Sun et al., 2021; Pamfil et al., 2020), multi-task learning (Chen et al., 2021), multi-domain settings (Zeng et al., 2021), federated learning (Ng & Zhang, 2022; Gao et al., 2023), and representation learning (Yang et al., 2021). With the growing interest in continuous structure learning methods (Vowels et al., 2022), a variety of theoretical and empirical studies have emerged. For instance, Ng et al. (2020) investigated the optimality conditions and convergence properties of continuously constrained approaches such as Zheng et al. (2018). In the bivariate case, Deng et al. (2023b) demonstrated that a suitable optimization strategy converges to the global minimum of the least squares objective. Zhang et al. (2022) and Bello et al. (2022) then identified potential gradient vanishing issues with existing DAG constraints (Zheng et al., 2018) and proposed adjustments to overcome these challenges.

**Contributions.** In this work, we focus on the graphical models represented as Directed Acyclic Graphs (DAGs). Our main contributions can be summarized as follows:

1. **Problem reformulation:** We introduce a new reformulation (9) of the discrete optimization problem for finding DAG as a stochastic optimization problem, and we discuss its properties in detail in Section 3.1. We demonstrate that the solution of this reformulated problem recovers the true DAG (Section 3.1).

2. **Novel algorithm:** Leveraging insights from stochastic optimization, we present a new framework (Algorithm 1) for DAG learning (Section 4) and present a simple yet effective algorithm $\psi$DAG (Algorithm 2) within the framework.

3. **Experimental comparison:** In Section 5, we demonstrate that the method $\psi$DAG scales very well with graph size, handling up to 10000 nodes. At that scale, the primary limitation is not computation complexity but the memory required to store the DAG itself. As a baseline, we compare $\psi$DAG with established DAG learning methods, including NOTEARS (Zheng et al., 2018), GOLEM (Ng et al., 2020), NOCURL (Yu et al., 2021) and DAGMA (Bello et al., 2022). We show a significant improvement in scalability, as baseline methods struggle with larger graphs. Specifically, NOTEARS (Zheng et al., 2018), GOLEM (Ng et al., 2020), NOCURL (Yu et al., 2021) and DAGMA (Bello et al., 2022) require more than 100 hours for graphs with over 3000 nodes, exceeding the allotted time.

## 2 BACKGROUND

In this section, we introduce the necessary graph notation and formalize the linear Structural Equation Model (SEM) framework used for learning Directed Acyclic Graphs (DAGs). For a detailed discussion of related methods and further literature, please refer to Appendix A.

### 2.1 GRAPH NOTATION

Let $\mathcal{G} \overset{def}{=} (V, E, w)$ represent a weighted directed graph, where $V$ denotes the set of vertices with cardinality $d \overset{def}{=} |V|$, $E \in 2^{V \times V}$ is the set of edges, and $w : V \times V \to \mathbb{R} \setminus \{0\}$ assigns weights to

the edges. The *adjacency matrix* $\mathbf{A}(\mathcal{G}) : \mathbb{R}^{d \times d}$ is defined such that $[\mathbf{A}(\mathcal{G})]_{ij} = 1$ if $(i, j) \in E$ and $0$ otherwise. Similarly, the *weighted adjacency matrix* $\mathbf{W}(\mathcal{G})$ is defined by $[\mathbf{W}(\mathcal{G})]_{ij} = w(i, j)$ if $(i, j) \in E$ and $0$ otherwise.

When the weight function $w$ is binary, we simplify the notation to $\mathcal{G} \stackrel{def}{=} (V, E)$. Similarly, when the graph $\mathcal{G}$ is clear from context, we shorthand the notation to $\mathbf{A} \stackrel{def}{=} \mathbf{A}(\mathcal{G})$ and $\mathbf{W} \stackrel{def}{=} \mathbf{W}(\mathcal{G})$.

We denote the space of DAGs as $\mathbb{D}$. Since we will be utilizing topological sorting of DAGs[1], we also denote the space of vertex permutations $\Pi$.

## 2.2 LINEAR DAG AND SEM

A Directed Acyclic Graph (DAG) model, defined on a set of $n$ random vectors $\mathbf{X} \in \mathbb{R}^{n \times d}$, where $\mathbf{X} \stackrel{def}{=} (X_1, \ldots, X_n)$ and $X_i \in \mathbb{R}^d$, consists of two components:

1. A DAG $\mathcal{G} = (V, E)$, which encodes a set of conditional independence relationships among the variables.
2. The joint distribution $P(\mathbf{X})$ with density $p(x)$, which is Markov with respect to the DAG $\mathcal{G}$ and factors as $p(x) = \prod_{i=1}^{d} p(x_i \mid x_{\mathrm{PA}_{\mathcal{G}}(i)})$, where $\mathrm{PA}_{\mathcal{G}}(i) = \{j \in V : X_j \rightarrow X_i \in E\}$ represents the set of parents of $X_i$ in $\mathcal{G}$.

This work focuses on the linear DAG model, which can be equivalently represented by a set of linear Structural Equation Models (SEMs). In matrix notation, the linear DAG model can be expressed as

$$\mathbf{X} = \mathbf{X}\mathbf{W} + \mathbf{N}, \tag{1}$$

where $\mathbf{W} = [\mathbf{W}_1 | \cdots | \mathbf{W}_d]$ is a weighted adjacency matrix, and $\mathbf{N} \stackrel{def}{=} (N_1, \ldots, N_n)$ is a matrix where each $N_i \in \mathbb{R}^d$ represents a noise vector with independent components. The structure of graph $\mathcal{G}$ is determined by the non-zero coefficients in $\mathbf{W}$; specifically $X_j \rightarrow X_i \in E$ if and only if the corresponding coefficient in $\mathbf{W}_i$ for $X_j$ is non-zero. The classical objective function is based on the least squares loss applied to the linear DAG model,

$$l(\mathbf{W}; \mathbf{X}) \stackrel{def}{=} \frac{1}{2n} \|\mathbf{X} - \mathbf{X}\mathbf{W}\|_F^2. \tag{2}$$

## 3 STOCHASTIC APPROXIMATION FOR DAGS

Our framework is built on a reformulation of the objective function as a stochastic optimization problem, aiming to minimize the stochastic function $F(w)$,

$$\min_{w \in \mathbb{R}^d} \left\{ F(w) \stackrel{def}{=} \mathbb{E}_\xi \left[ f(w, \xi) \right] \right\}, \tag{3}$$

where $\xi \in \Xi$ is a random variable that follows the distribution $\Xi$. This formulation is common in stochastic optimization where computing the exact expectation is infeasible, but the values of $f(w, \xi)$ and its stochastic gradients $g(w, \xi)$ can be computed. Linear and logistic regressions are classical examples of such problems.

To address this problem, two main approaches exist: Stochastic Approximation (SA) and Sample Average Approximation (SAA). The SAA approach involves sampling a fixed number $n$ of random variables or data points $\xi_i$ and then minimizing their average $\tilde{F}(w)$:

$$\min_{w \in \mathbb{R}^d} \left\{ \tilde{F}(w) \stackrel{def}{=} \frac{1}{n} \sum_{i=1}^{n} f(w, \xi_i) \right\}. \tag{4}$$

Now, the problem in (4) becomes deterministic and can be solved using various optimization methods, such as gradient descent. However, the main drawback of this approach is that the solution to (4) $\tilde{w}^*$ is not necessarily equal to the solution of the original problem in (3). Even with a perfect solution of (4), there will still be a gap $\|\tilde{w}^* - w^*\| = \delta_x$ and $F(\tilde{w}^*) - F^* = \delta_F$ between approximate and true

---

[1]Topological sorting of a graph $\mathcal{G} \stackrel{def}{=} (V, E, w)$ refers to vertex ordering $V_1, V_2, \ldots, V_d$ such that $E$ contains no edges of the form $V_i \rightarrow V_j$, where $i \leq j$. Importantly, every DAG has at least one topological sorting.

---

**Algorithm 1** $\psi$DAG framework

---

1: **Requires:** Initial model $\mathbf{W}_0 \in \mathbb{R}^{d \times d}$, such that $\text{diag}(\mathbf{W}_0) = 0$.
2: **for** $k = 0, 1, 2 \ldots, K - 1$ **do**
3: $\quad \mathbf{W}_k^{(1/3)} = \mathcal{A}_1(\mathbf{W}_k)$ $\qquad\qquad\qquad\qquad\qquad\qquad\qquad$ $\{\mathbf{W}_k^{(1/3)} \in \mathbb{R}^{d \times d}\}$
4: $\quad (\mathbf{W}_k^{(2/3)}, \pi_k) = \psi(\mathbf{W}_k^{(1/3)})$ $\qquad\qquad\qquad\qquad\qquad\quad$ $\{\mathbf{W}_k^{(2/3)} \in \mathbb{D}\}$
5: $\quad \mathbf{W}_{k+1} = \mathcal{A}_2(\mathbf{W}_k^{(2/3)}; \pi_k)$ $\qquad\qquad\qquad\qquad\quad$ $\{\mathbf{W}_{k+1} \in \mathbb{D} \subset \mathbb{R}^{d \times d}\}$
6: **end for**
7: **Output:** $\mathbf{W}_K$.

---

solution. These gaps are dependent on the sample size $n$.

Stochastic Approximation (SA) minimizes the true function $F(w)$ by utilizing the stochastic gradient $g(w, \xi)$. Below, we provide the formal definition of a stochastic gradient.

**Assumption 1.** *For all $w \in \mathbb{R}^d$, we assume that stochastic gradients $g(w, \xi) \in \mathbb{R}^d$ satisfy*

$$\mathbb{E}[g(w, \xi) \mid w] = \nabla F(w), \tag{5}$$

$$\mathbb{E}\left[\|g(w, \xi) - \nabla F(w)\|^2 \mid w\right] \le \sigma_1^2. \tag{6}$$

We use these stochastic gradients in SGD-type methods:

$$w_{t+1} = w_t - h_t g(w_t, \xi_i), \tag{7}$$

where $h_t$ is a step-size schedule. SA originated with the pioneering paper by Robbins & Monro (1951). For convex and $L$-smooth function $F(w)$, Polyak (1990); Polyak & Juditsky (1992); Nemirovski et al. (2009); Nemirovski & Yudin (1983) developed significant improvements to SA method in the form of longer step-sizes with iterate averaging, and obtained the convergence guarantee

$$\mathbb{E}\left[F(w_T) - F(x^*)\right] \le \mathcal{O}\left(\frac{\sigma_1 R}{\sqrt{T}} + \frac{L_1 R^2}{T}\right).$$

Lan (2012) developed an optimal method with a guaranteed convergence rate $\mathcal{O}\left(\frac{\sigma_1 R}{\sqrt{T}} + \frac{L_1 R^2}{T^2}\right)$, matching the worst-case lower bounds. The key advantage of SA is that it provides convergence guarantees for the original problem in (3). Additionally, methods effective for the SA approach tend to perform well for the SAA approach as well.

### 3.1 Stochastic Reformulation

Using the perspective of Stochastic Approximation, we can rewrite the linear DAG in (1) as

$$x = X_i = \left[\mathbf{I} - \mathbf{W}_*^\top\right]^{-1} N_i, \tag{8}$$

where $\mathbf{W}^*$ is a true DAG that corresponds to the full distribution, and our goal is to find a DAG $\mathbf{W}$ that is close to $\mathbf{W}^*$. If we assume that $x = X_i$ is a random vector sampled from a distribution $\mathcal{D}$, we can express the objective function as an expectation,

$$\min_{\mathbf{W} \in \mathbb{D}} \mathbb{E}_{x \sim \mathcal{D}} \left[l(\mathbf{W}; x) \overset{def}{=} \tfrac{1}{2}\|x - \mathbf{W}^\top x\|^2\right]. \tag{9}$$

For $x$ from (8) we can calculate $\|x - \mathbf{W}^\top x\| = \|(\mathbf{I} - \mathbf{W}^\top)x\| = \|(\mathbf{I} - \mathbf{W}^\top)\left[\mathbf{I} - \mathbf{W}_*^\top\right]^{-1} N_i\|$, which implies that the minimizer of (9) recovers the true DAG. Conversely, this is not the case for methods such as Zheng et al. (2018), Ng et al. (2020), and Bello et al. (2022), which are based on SAA approaches using the loss functions defined in (2), (11), (12), and (13).

## 4 Scalable Optimization Framework for DAG Learning

In this section, we present our proposed scalable optimization framework for DAG learning. We begin by showing that using a fixed vertex ordering can lead to suboptimal solutions, as demonstrated in Section 4.1. Motivated by this, we develop a three-stage framework that alternates between

---

**Algorithm 2** $\psi$DAG

---

1: **Requires:** initial model $\mathbf{W}_0 \in \mathbb{R}^{d \times d}$, numbers or iterations $\tau_1, \tau_2$.
2: **for** $k = 0, 1, 2 \ldots, K - 1$ **do**
3:     $\mathbf{W}_k^{(1/3)} = \mathsf{SGD}(\mathbf{W}_k)$                         $\{\tau_1 \text{ iterations over } \mathbb{R}^{d \times d}\}$
4:     $(\mathbf{W}_k^{(2/3)}, \pi_k) = \text{Algorithm 3 } (\mathbf{W}_k^{(1/3)})$
5:     $\mathbf{W}_{k+1} = \mathsf{SGD}_{\pi_k}(\mathbf{W}_k)$             $\{\tau_2 \text{ iterations preserving ordering } \pi_k\}$
6: **end for**
7: **Output:** $\mathbf{W}_K$

---

unconstrained optimization, projection onto the DAG space, and constrained optimization guided by topological ordering.

Instead of strictly enforcing DAG constraints throughout the entire iteration process, we propose a novel, scalable optimization framework that consists of three main steps:

1. Running an optimization algorithm $\mathcal{A}_1$ without any DAG constraints, only forcing the diagonal to be zero ($\mathrm{diag}(W_k) = 0$), $\mathcal{A}_1 : \mathbb{R}^{d \times d} \to \mathbb{R}^{d \times d}$.
2. Finding a DAG that is close to the current iterate using a projection $\psi : \mathbb{R}^{d \times d} \to (\mathbb{D}, \Pi)$, which also returns its topological sorting $\pi$.
3. Running the optimization algorithm $\mathcal{A}_2$ while preserving the vertex order, $\mathcal{A}_2 : (\mathbb{D}; \Pi) \to \mathbb{D}$.

This design enables efficient and accurate structure learning while avoiding the computational burden of enforcing DAG constraints at every iteration.

### 4.1 OPTIMIZATION FOR THE FIXED VERTEX ORDERING

Let us clarify how to optimize while preserving the order of the vertices in step 3 of the framework. Given a DAG $\mathcal{G}$, we can construct its topological ordering, denoted as $ord(\mathcal{G})$. In this ordering, for every edge, the start vertex appears earlier in the sequence than the end vertex. In general, this ordering is not unique. In the space of DAGs with $d$ vertices $\mathbb{D}$, there are $d!$ possible topological orderings.

Once we have a topological ordering of the DAG, we can construct a larger DAG, $\hat{\mathcal{G}}$, by performing the transitive closure of $\mathcal{G}$. This new DAG $\hat{\mathcal{G}}$ contains all the edges of the original DAG, and additionally, it includes an edge between vertices $V_i$ and $V_j$ if there exists the path from $V_i$ to $V_j$ in $\mathcal{G}$. Thus, $\hat{\mathcal{G}}$ is an expanded version of $\mathcal{G}$.

Now, the question arises: is it possible to construct an even larger DAG that contains both $\mathcal{G}$ and $\hat{\mathcal{G}}$? The answer is yes! We call this graph the *Full DAG*, denoted by $\tilde{\mathcal{G}}$, which is constructed via full transitive closure[2]. In $\tilde{\mathcal{G}}$, there is an edge from vertex $V_i$ to vertex $V_j$ if $i < j$ is in topological order $ord(\mathcal{G})$. This makes $\tilde{\mathcal{G}}$ the maximal DAG that includes $\mathcal{G}$. Note that for every topological sort, there is a corresponding full DAG. So, there are a total of $d!$ different full DAGs in the space of DAGs with $d$ vertices $\mathbb{D}$.

We are now ready to discuss the optimization part. Let us formulate the following optimization problem

$$\min_{\mathbf{W} \in \mathbb{R}^{d \times d}} \mathbb{E}_{x \sim \mathcal{D}} \left[ l(\mathbf{W} \cdot \mathbf{A}; x) = \tfrac{1}{2} \| x - (\mathbf{W} \cdot \mathbf{A})^\top x \|^2 \right], \tag{10}$$

where $(\cdot)$ denotes elementwise matrix multiplication. In this formulation, $\mathbf{A}$ acts as a mask, specifying coordinates that do not require gradient computation. The problem in (10) is a quadratic convex stochastic optimization problem, which can be efficiently solved using stochastic gradient descent (SGD)-type methods. These methods guarantee convergence to the global minimum, with a rate of $\mathcal{O}\left( \frac{\sigma_1 R}{\sqrt{T}} + \frac{L_1 R^2}{T} \right)$.

---

[2]Informally, for set of edges $E$, the transitive closure $E^+$ is the smallest set that includes edges $(a, b)$ whenever there is a path from $a$ to $b$ within $E$. Note that $E^+$ is the smallest superset of $E$ that satisfies $(a, c) \in E^+$ whenever $(a, b) \in E^+, (b, c) \in E^+$.

---

**Algorithm 3** Projection $\psi(\mathbf{W})$ computing the "closest" vertex ordering (recursive form)

---

1: **Requires:** Model $\mathbf{W} \in \mathbb{R}^{d \times d}$, (optional) weights $\mathbf{L} \in \mathbb{R}^{d \times d}$ with default value $\mathbf{L} = \mathbf{1}\mathbf{1}^\top$.
2: **for** $k = 1, \ldots, d$ **do**
3:      Set $r_k = \| (\mathbf{W} \circ \mathbf{L}) [k][:] \|^2$
4:      Set $c_k = \| (\mathbf{W} \circ \mathbf{L}) [:][k] \|^2$
5: **end for**
6: Set $i_c = \arg \min_{k \in \{1, \ldots, d\}} c_k$
7: Set $i_r = \arg \min_{k \in \{1, \ldots, d\}} r_k$
8: **if** $r_{i_r} <= c_{i_c}$ **then**
9:      **Output:** $[\psi(\mathbf{W}(i_c, i_c), \mathbf{L}(i_c, i_c)), i_r]$
10: **else**
11:      **Output:** $[i_c, \psi(\mathbf{W}(i_c, i_c), \mathbf{L}(i_c, i_c))]$
12: **end if**
     {By $A(i, j)$ we denote the submatrix $A[1, \ldots, i-1, i+1, \ldots, d][1, \ldots, j-1, j+1, \ldots, d]$}

---

Assume that $\mathcal{G}^*$ is the true DAG with a weighted adjacency matrix $\mathbf{W}^*$, which is the solution we aim to find. Next, we can have the true ordering $ord(\mathcal{G}^*)$ and the true full DAG $\tilde{\mathcal{G}}^*$ with its adjacency matrix $\mathbf{A}(\tilde{\mathcal{G}}^*)$. The optimization problem in (9), with the solution $\mathbf{W}^*$, can be addressed by solving the optimization problem in (10) with $\mathbf{A} = \mathbf{A}(\mathcal{G}^*)$. This result indicates that if we know the true topological ordering $ord(\mathcal{G}^*)$, then we can recover the true DAG $\mathbf{W}^*$ with high accuracy. From a discrete optimization perspective, this approach significantly reduces the space of constraints from $2^{d^2 - d}$ to $d!$.

To illustrate the specificity of the minimizer of the proposed problem, Figure 2 demonstrates that minimizing (9) over a fixed random vertex ordering does not approach the true solution of (9). The "Correct order" curve demonstrates the convergence of (10) when the true ordering $ord(\mathcal{G}^*)$ is known.

Note that for a fixed vertex ordering and fixed adjacency matrix $\mathbf{A}$, the objective in (10) becomes separable, enabling parallel computation for large-scale problems. In this work, we solved the minimization problem in (10) for the number of nodes up to $d = 10^4$, at which point the limiting factor was the memory to store $\mathbf{W} \in \mathbb{R}^{d \times d}$. Through parallelization and efficient memory management, it is possible to solve even larger problems.

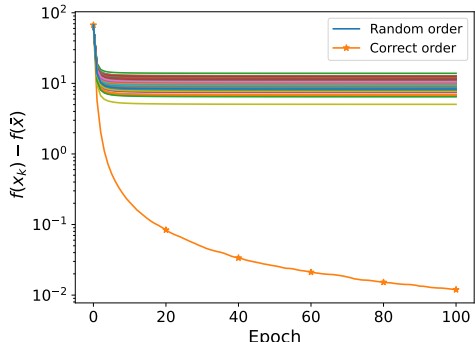

Figure 2: Minimizing (9) using SGD over a fixed topological ordering on ER4 with $d = 100$ and Gaussian noise.

## 4.2 METHODOLOGY

We now introduce the method $\psi\mathsf{DAG}$, which implements the framework outlined in Algorithm 1.

For simplicity, we select algorithm $\mathcal{A}_1$ as $\tau_1$ steps of Stochastic Gradient Descent (SGD). Similarly, $\mathcal{A}_2$ consists of $\tau_2$ steps SGD, where gradients are projected onto the space spanned by DAG's topological sorting, thus preserving the vertex order. It is important to reiterate that SGD is guaranteed to converge to the neighborhood of the solution. In the implementation, we employed an advanced version of SGD, Universal Stochastic Gradient Method from Rodomanov et al. (2024).

The implementation of the projection method is simple as well. We compute a "closest" topological sorting and remove all edges not permitted by this ordering. The topological sorting is computed by a heuristic that calculates norms of all rows and columns to find the lowest value $v_i$. The corresponding vertex $i$ is then assigned to the ordering based on the following rule:

- If $v_i$ was the column norm, $i$ is assigned to the beginning of the ordering.
- If $v_i$ was the row norm, $i$ is assigned to the end of the ordering.

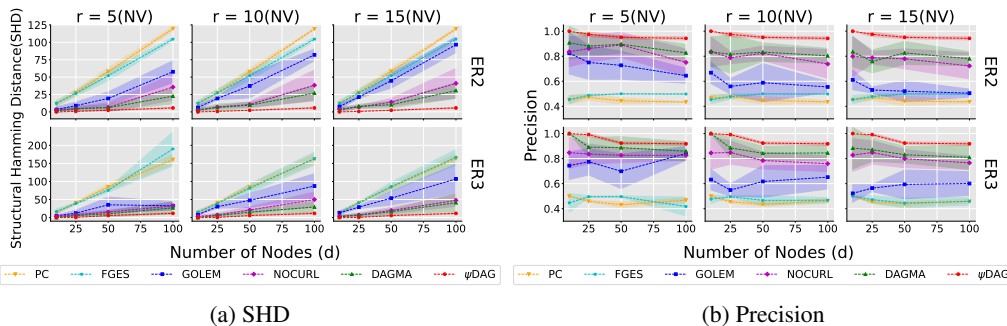

Figure 3: Structure recovery performance under Gaussian noise with non-equal variances (NV) across varying noise ratios ($r \in \{5, 10, 15\}$). Rows correspond to different random graph types, and columns represent increasing noise ratios. Metrics include Structural Hamming Distance (SHD $\downarrow$) and Precision ($\uparrow$). We report the mean values, and the standard error is indicated by shaded regions.

This step reduces the number of vertices, and the remaining vertices are topologically sorted using a recursive call. We formalize this procedure in Algorithm 3. Note that this procedure can be efficiently implemented without recursion and with the computation cost $\mathcal{O}(d^2)$.

## 5 EXPERIMENTS

We experimentally compare our method, $\psi$DAG[3], with several baselines including PC (Ramsey et al., 2012), FGES (Meek, 1997; Chickering, 2002), NOTEARS (Zheng et al., 2018), GOLEM (Ng et al., 2020), NOCURL (Yu et al., 2021) and DAGMA (Bello et al., 2022). As it is established that DAGMA Bello et al. (2022) is an improvement over NOTEARS Zheng et al. (2018), we use mostly the former in our experiments. To ensure a fair comparison, we avoid extensive hyperparameter tuning across all baseline methods. Specifically, we apply the same thresholding procedure as used in Zheng et al. (2018), Ng et al. (2020), Yu et al. (2021), and Bello et al. (2022) across all scenarios.

### 5.1 SYNTHETIC DATA GENERATION

We generate ground truth DAGs with $d$ nodes and an average of $k \times d$ edges, where $k \in \{2, 3, 4, 6\}$ is a sparsity parameter. The graph structure is based on either the Erdős-Rényi (ER) or the Scale-Free (SF) models. Together with the sparsity level, we denote the graphs as ER$k$ or SF$k$, respectively. Each edge is assigned a random weight uniformly sampled from the interval $[-2, -0.5] \cup [0.5, 2]$ following the standard practice used in previous work (Zheng et al., 2018; Ng et al., 2020; Yu et al., 2021; Bello et al., 2022) to ensure consistency across methods.

Following the linear Structural Equation Model (SEM), we generate the observed data $\mathbf{X} \in \mathbb{R}^{n \times d}$ using $\mathbf{X} = \mathbf{N}(\mathbf{I} - \mathbf{W})^{-1}$, where $\mathbf{N}$ consists of $n$ independent and identically distributed (i.i.d.) noise samples drawn from Gaussian, exponential, or Gumbel distributions. We evaluated both equal-variance (EV) and non-equal variance (NV) Gaussian noise settings. In the EV case, the noise for all variables is scaled by a constant factor of 1.0. In the NV setting, the noise variables have heterogeneous variances. We randomly choose two variables, one is fixed to have a variance of 1, and the other is fixed to have a variance $r \in \{5, 10, 15\}$. The remaining variables are assigned variances drawn uniformly at random from the interval $[1, r]$. This setup enables evaluation of robustness under noise heterogeneity. For further details, we refer the reader to Ng et al. (2024). A visual comparison under both EV and NV (with $r = 5$) is shown in Figure 1, highlighting the robustness of $\psi$DAG to non-uniform noise levels. A more detailed description can be found in the Appendix C.

---

[3]Implementation of the proposed algorithm is available at https://anonymous.4open.science/r/psiDAG-8F42. We use the Universal Stochastic Gradient Method (Rodomanov et al., 2024) as the inner optimizer.

## 5.2 STRUCTURE RECOVERY

We evaluate the structure learning capabilities of our method, as shown in Figure 3, using synthetic data generated from ER2 and ER3 graphs with varying node counts $d \in \{10, 25, 50, 100\}$. For brevity, we report only the results for Structural Hamming Distance (SHD) and precision across different noise ratios $r \in \{5, 10, 15\}$ in the non-equal variance (NV) Gaussian setting. The complete results for additional metrics, including the F1 score and the recall, are given in the Appendix D.1. Lower SHD and higher precision, F1 score, and recall values indicate better structure recovery. We compare against several representative baselines, including PC, FGES, NOCURL, GOLEM, and DAGMA.

Consistent with prior work, all methods perform well in terms of SHD when the number of nodes $d$ and the noise ratio $r$ are small. However, the performance of FGES and PC deteriorates rapidly, even for a moderate number of nodes such as $d = 50$, with SHD increasing significantly. In contrast, our method maintains low SHD across all settings and consistently outperforms baselines as noise heterogeneity increases. Figure 4 shows that the SHD of $\psi$DAG remains stable even as $r$ increases from 5 to 1024, further emphasizing its stability and robustness. Meanwhile, DAGMA fails to converge for $r > 15$, limiting its applicability in high-noise regimes.

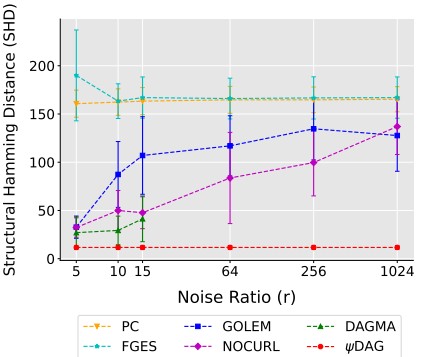

Figure 4: Effect of noise ratio on SHD for ER3 graphs with $d = 100$ in the non-equal variance (NV) Gaussian setting. Error bars denote standard deviation over 3 random seeds. $\psi$DAG remains stable as $r$ increases, while other methods degrade and DAGMA fails to converge for $r > 15$.

## 5.3 SCALABILITY COMPARISON

We assess the scalability of the proposed algorithm, $\psi$DAG, by comparing its runtime against GOLEM, NOCURL, and DAGMA. All methods are run until the objective function converges close to the solution, $f(x_k) - f(\overline{x}) \leq 0.1 \cdot f(\overline{x})$. Figure 5 reports runtime comparisons for ER2 and ER4 graphs under Gaussian, Exponential, and Gumbel noise for graph sizes $d \in \{10, 50, 100, 500, 1000\}$. Due to space constraints, additional results on larger graphs (up to $d = 10,000$) are presented in Appendix D.2, where Figure 9 further highlights the efficiency of $\psi$DAG in high-dimensional settings.

Across all scenarios, $\psi$DAG demonstrates consistently lower runtime compared to baselines, particularly as graph size and density increase. While DAGMA is marginally faster than $\psi$DAG on very small and sparse graphs ($d < 100$), the gap closes quickly with larger graphs. For $d > 100$, $\psi$DAG consistently exhibits superior runtime performance across both sparse and dense graph types. On

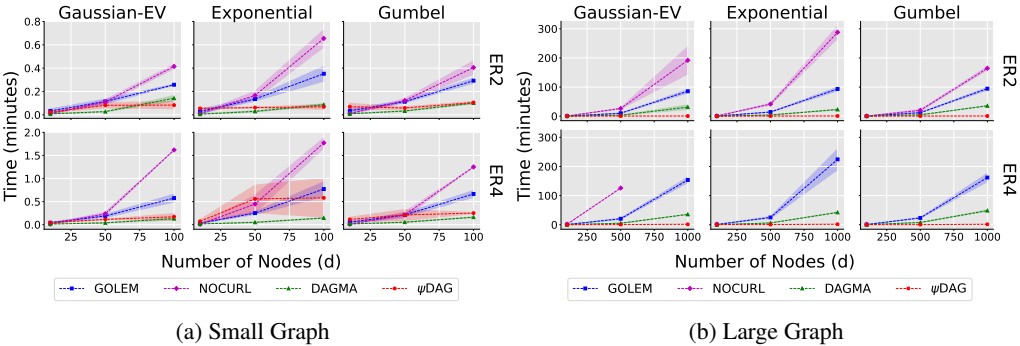

(a) Small Graph        (b) Large Graph

Figure 5: Runtime (minutes) of GOLEM, NOCURL, DAGMA, and $\psi$DAG on ER2 and ER4 graphs with increasing number of nodes $d \in \{10, 50, 100, 500, 1000\}$. The columns correspond to different noise distributions: Gaussian (left), exponential (middle), and Gumbel (right). Figure 5a shows results for small graphs ($d \leq 100$) and 5b for large graphs ($d > 100$). $\psi$DAG demonstrates significantly better scalability as the number of nodes increases.

sparse graphs, it converges reliably within a few hours, even at $d = 10,000$, whereas GOLEM and NOCURL exceed a 36-hour runtime for $d \geq 3000$, and DAGMA does so for $d \geq 5000$.

Furthermore, we observe that several baselines fail to meet the convergence criterion even for smaller graphs. For instance, NOCURL does not converge for ER4 graphs with Gaussian-EV noise when $d > 500$, and it fails completely for Exponential and Gumbel noise when $d > 100$. In the ER6 graph, the three baselines GOLEM, NOCURL, and DAGMA do not converge in at least one of the three random seeds. Non-converging runs are excluded from reported statistics. In contrast, $\psi$DAG converges in all runs and maintains competitive runtime performance even at large scale, underscoring both its robustness and practical efficiency.

### 5.4 REAL-WORLD EXPERIMENT

We further evaluate $\psi$DAG on a widely used real-world dataset, the *causal protein signaling network*, from Sachs et al. (2005b) and compare it with NOTEARS (Zheng et al., 2018), GOLEM (Ng et al., 2020), NOCURL (Yu et al., 2021), and DAGMA (Bello et al., 2022). This dataset captures the expression levels of proteins and phospholipids in human cells under various experimental conditions. It has been extensively used in the literature on causal discovery due to its well-established ground truth and biological relevance. The dataset consists of $n = 853$ observational samples and $d = 11$ variables, with a ground truth DAG containing 17 edges. Despite its small size, it remains a challenging benchmark for causal structure learning algorithms (Zheng et al., 2018; Ng et al., 2020; Gao et al., 2021). We follow the common evaluation setup and apply a threshold of 0.3 across all methods for a fair comparison.

As shown in Table 1, $\psi$DAG achieves superior performance across all metrics: lower Structural Hamming Distance (SHD), higher True Positive Rate (TPR), and lower False Positive Rate (FPR). A more detailed description can be found in Appendix C. We omit the results for DAGMA as it fails to converge on this dataset: its solution $\mathbf{W}$ diverges from the feasible domain in the very first iteration.

Table 1: Performance of top methods on the protein signaling dataset (Sachs et al., 2005b).

|  | SHD↓ | TPR↑ | FPR↓ |
|---|---|---|---|
| NOTEARS (Zheng et al., 2018) | 15 | 0.29 | 0.26 |
| GOLEM (Ng et al., 2020) | 26 | 0.29 | 0.47 |
| NOCURL (Yu et al., 2021) | 22 | 0.35 | 0.45 |
| $\psi$DAG (Alg.2) | **14** | **0.41** | **0.18** |

## 6 CONCLUSION

We introduce a novel framework for learning Directed Acyclic Graphs (DAGs) that addresses the scalability and computational challenges of existing methods. Our approach leverages Stochastic Approximation techniques in combination with Stochastic Gradient Descent (SGD)-based methods, allowing for efficient optimization even in high-dimensional settings. A key contribution of our framework is the introduction of new projection techniques that effectively enforce DAG constraints, ensuring that the learned structure adheres to the acyclicity requirement without the need for computationally expensive penalties or constraints seen in prior works.

The proposed framework is theoretically grounded, with convergence guarantees to a feasible local minimum. One of its main advantages is its low iteration complexity, making it highly suitable for large-scale structure learning problems, where traditional methods often struggle with runtime and memory limitations. Through extensive experiments, we show that our approach consistently outperforms strong baselines including NOTEARS (Zheng et al., 2018), GOLEM (Ng et al., 2020), NOCURL (Yu et al., 2021), and DAGMA (Bello et al., 2022) in both runtime and structure recovery accuracy. Notably, our method demonstrates robust performance in settings with high noise heterogeneity and varying graph densities.

**Limitations and Future Work.** In this paper, we have focused on presenting a novel framework for differentiable DAG learning, which integrates a stochastic approach to achieve computational efficiency. While the current results are focused on linear SEMs for simplicity, extending the proposed algorithm to handle nonlinear SEMs (Zheng et al., 2020) is a natural direction for future work. Exploring variance reduction optimization methods is another promising path.

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

CONTENTS

## A  RELATED WORK

A significant body of research on DAG learning revolves around non-convex continuous optimization frameworks, such as NOTEARS (Zheng et al., 2018), GOLEM (Ng et al., 2020), NOCURL (Yu et al., 2021), and DAGMA (Bello et al., 2022). These approaches address the DAG constraint using either smooth approximations or novel penalty functions, but they are often computationally expensive and lack scalability.

Zheng et al. (2018) addressed the constrained optimization problem

$$\min_{\mathbf{W} \in \mathbb{R}^{d \times d}} \ell(\mathbf{W}; \mathbf{X})_{\text{NOTEARS}} \stackrel{def}{=} \frac{1}{2n} \|\mathbf{X} - \mathbf{X}\mathbf{W}\|_F^2 + \lambda \|\mathbf{W}\|_1 \quad \text{subject to} \quad h(\mathbf{W}) = 0, \quad (11)$$

where $\ell(\mathbf{W}; \mathbf{X})$ represents the least squares objective and $h(\mathbf{W}) := \text{tr}(e^{\mathbf{W} \odot \mathbf{W}}) - d$ enforces the DAG constraint. Additionally, an $\ell_1$ regularization term $\lambda \|\mathbf{W}\|_1$, where $\|\cdot\|_1$ is the element-wise $\ell_1$-norm and $\lambda$ is a hyperparameter incorporated into the objective function. This formulation addresses the linear case with equal noise variances, as discussed in Loh & Bühlmann (2014) and Peters & Bühlmann (2014). This constrained optimization problem is solved using the augmented Lagrangian method (Bertsekas et al., 1999), followed by thresholding the obtained edge weights. However, since this approach computes the acyclicity function via the matrix exponential, each iteration incurs a computational complexity of $\mathcal{O}(d^3)$, which significantly limits the scalability of the method.

Ng et al. (2020) introduced the GOLEM method, which enhances the scoring function by incorporating an additional log-determinant term, $\log |\det(\mathbf{I} - \mathbf{W})|$ to align with the Gaussian log-likelihood,

$$\min_{\mathbf{W} \in \mathbb{R}^{d \times d}} \ell(\mathbf{W}; \mathbf{X})_{\text{GOLEM}} \stackrel{def}{=} \frac{d}{2} \log \|\mathbf{X} - \mathbf{X}\mathbf{W}\|_F^2 - \log |\det(\mathbf{I} - \mathbf{W})| + \lambda_1 \|\mathbf{W}\|_1 + \lambda_2 h(\mathbf{W}), \quad (12)$$

where $\lambda_1$ and $\lambda_2$ serve as regularization hyperparameters within the objective function. Although the newly added log-determinant term is zero when the current model $\mathbf{W}$ is a DAG, this score function does not provide an exact characterization of acyclicity. Specifically, the condition $\log |\det(\mathbf{I} - \mathbf{W})| = 0$ does not imply that $\mathbf{W}$ represents a DAG.

Bello et al. (2022) introduces a novel acyclity characterization for DAGs using a log-determinant function,

$$\min_{\mathbf{W} \in \mathbb{R}^{d \times d}} \ell(\mathbf{W}; \mathbf{X})_{\text{DAGMA}} \stackrel{def}{=} \frac{1}{2n} \|\mathbf{X} - \mathbf{X}\mathbf{W}\|_F^2 + \lambda_1 \|\mathbf{W}\|_1 \quad \text{subject to} \quad h_{ldet}^s(\mathbf{W}) = 0, \quad (13)$$

where $h_{ldet}^s(\mathbf{W}) \stackrel{def}{=} -\log \det(s\mathbf{I} - \mathbf{W} \circ \mathbf{W}) + d \log s$, and it is both exact and differentiable.

In practice, the augmented Lagrangian method enforces the hard DAG constraint by increasing the penalty coefficient toward infinity, which requires careful parameter fine-tuning and can lead to numerical difficulties and ill-conditioning (Birgin et al., 2005; Ng et al., 2022a). As a result, existing methods face challenges in several aspects of optimization, including careful selection of constraints, high computational complexity, and scalability issues.

Yu et al. (2021) introduce a novel formulation for DAG structure learning by expressing the weighted adjacency matrix as the Hadamard product of a skew-symmetric matrix and the gradient of a potential function on graph nodes. This representation avoids explicit acyclicity constraints and enables a continuous, constraint-free optimization framework. However, although NOCURL avoids direct constraints through this parameterization and Hodge decomposition, it still relies on repeated L-BFGS (Broyden, 1967) optimization steps , which can become computationally expensive for large graphs. In contrast, $\psi$DAG avoids both acyclicity constraints and expensive optimization procedures, allowing for more efficient scaling in high-dimensional settings.

Other works, such as Chen et al. (2019), proposed variance ordering procedures for estimating topological orderings under equal error variances. Although these methods naturally extend to high-dimensional settings, their reliance on controlling the maximum in-degree of the graph becomes computationally intensive as graph density increases. In contrast, $\psi$DAG avoids these assumptions and demonstrates scalability on graphs with up to $10,000$ nodes. Gao et al. (2022b) focused on theoretical guarantees for Gaussian DAG models, obtaining minimax optimal bounds for structural recovery. Although their work offers valuable insights into sample efficiency, it does not address the computational challenges of large-scale DAG learning.

Wei et al. (2020) examined optimization challenges in NOTEARS by analyzing the KKT conditions and proposed the KKTS algorithm as a post-processing enhancement. While this method improves the structural Hamming distance (SHD), its reliance on specific constraints and post-hoc refinements limits its applicability. In contrast, $\psi$DAG reformulates DAG learning as a stochastic optimization problem, seamlessly integrating gradient-based methods for large-scale graphs.

Additionally, Deng et al. (2023a) introduced a bilevel algorithm that iteratively refines topological orders through node swaps, achieving local minima or KKT points. However, this approach is

constrained by a specific function $h(B) = \sum_{i=1}^{d} c_i \text{Tr}(B^i)$, which is computationally expensive and limits its scalability to applications that involve larger graphs. Consequently, their experiments are restricted to synthetic datasets with graphs containing up to $d = 100$ nodes. Moreover, the algorithm initializes the $\mathbf{W}$ matrix using linear regression coefficients in the least squares case, resulting in a different starting point for optimization, which makes direct comparisons with other methods challenging. Our method addresses these limitations by generalizing the DAG learning framework and demonstrating superior scalability and performance on both synthetic and real datasets.

Compared to permutation-based methods such as SP (Raskutti & Uhler, 2019), Efficient Permutation Discovery (Squires et al., 2020), and GRaSP (Lam et al., 2022), $\psi$DAG avoids exhaustive or greedy searches over permutations. Instead, it leverages a novel projection technique that efficiently infers causal orders without the computational overhead associated with permutation-based algorithms. This design choice allows $\psi$DAG to maintain accuracy while offering superior scalability and efficiency in learning DAG structures.

A recent permutation-based approach, BOSS (Andrews et al., 2023), performs greedy search over variable orderings and, for each ordering, constructs the DAG that optimizes a BIC score. This method effectively identifies a representative of the underlying Markov equivalence class but requires explicit exploration of orderings, which becomes computationally demanding for large graphs. In contrast, $\psi$DAG avoids permutation enumeration entirely as it operates in the continuous space of weighted adjacency matrices and enforces acyclicity via projection, enabling scalability to much larger problem sizes.

While many of these works focus on specific assumptions, penalty terms, or theoretical guarantees, our framework prioritizes scalability, flexibility, and applicability. To overcome these challenges, we propose a novel framework for enforcing the acyclicity constraint, utilizing a low-cost projection method. This approach significantly reduces iteration complexity and eliminates the need for expensive hyperparameter tuning.

## B  Theoretical Results

In this section, we present some theoretical properties of the DAG set and analyze the convergence of the proposed method.

**Lemma 2.** *The DAG set $\mathbb{D}$ is a conic set. Specifically, for any $\mathbf{W} \in \mathbb{D}$ and $\alpha \geq 0$, we have $\alpha\mathbf{W} \in \mathbb{D}$. Additionally, the DAG set $\mathbb{D}$ includes the entire line, meaning that for any $\mathbf{W} \in \mathbb{D}$ and $\alpha \in \mathbb{R}$, $\alpha\mathbf{W} \in \mathbb{D}$.*

*Proof.* We begin by observing that $\mathbf{0} \in \mathbb{D}$, as a graph with no edges is trivially a DAG. Next, consider any $\mathbf{W} \in \mathbb{D}$ and $\alpha \in \mathbb{R} \setminus \{0\}$. Scaling $\mathbf{W}$ by $\alpha$ does not alter the structure of the graph; it only changes the edge weights. Since the graph remains acyclic, $\alpha\mathbf{W} \in \mathbb{D}$. Thus, the DAG set $\mathbb{D}$ satisfies the stated properties. $\qquad\square$

Now, let us move to the subsets of DAG, which are based on a topological ordering $\pi$.

**Definition 3.** *A topological ordering $\pi$ of a directed graph is a linear ordering of its vertices such that, for every directed edge $(u, v)$ from vertex $u$ to vertex $v$, $u$ comes before $v$ in the ordering. We call $Ord(\mathbf{W})$ a set of all possible topological orderings for DAG $\mathbf{W}$ and $ord(\mathbf{W})$ is one of the orderings.*

For the graphs with $d$ vertices, there are exactly $d!$ distinct topological orderings.
Every topological ordering $\pi$ corresponds to subspace of all DAGs which can have this topological ordering, we call it $\pi$-subspace DAG.

**Definition 4.** *A $\pi$-subspace $\mathbb{D}_\pi$ is a set of all DAGs $\mathbf{W}$ such that $\pi \in Ord(\mathbf{W})$.*

Let us prove that $\pi$-subspace $\mathbb{D}_\pi$ is a linear subspace.

**Lemma 5.** *$\mathbb{D}_\pi$ is a linear subspace, meaning for any $\mathbf{W}_1 \in \mathbb{D}_\pi, \mathbf{W}_2 \in \mathbb{D}_\pi, \alpha \in \mathbb{R}, \beta \in \mathbb{R}$, $\mathbf{W} = \alpha\mathbf{W}_1 + \beta\mathbf{W}_2 \in \mathbb{D}_\pi$.*

*Proof.* We should simply note that any non-zero value in $\mathbf{W}_1$ corresponds to an edge between vertices $u$ and $v$ such that $v$ is after $u$ in the ordering $\pi$. The same holds for $\mathbf{W}_2$. Hence, any non-zero value in $\mathbf{W}$ holds the ordering $\pi$. $\square$

Next, we highlight that the DAG set $\mathbb{D}$ is a union of $\pi$-subspaces for all possible orderings $\pi$.

**Lemma 6.** *The DAG set $\mathbb{D}$ is a union of all $\pi$-subspaces.*

$$\mathbb{D} = \cup_\pi \mathbb{D}_\pi.$$

*Proof.* For any DAG $\mathbf{W} \in \mathbb{D}$ there exists a topological ordering $\pi$, hence $\mathbf{W} \in \mathbb{D}_\pi \in \cup_\pi \mathbb{D}_\pi$. On the other side, all elements of $\cup_\pi \mathbb{D}_\pi$ are DAGs by definition and belongs to $\mathbb{D}$.

$\square$

**Lemma 7** ($\mathbf{W}_*$ is a global minimizer of (9))**.** *Let $\mathbf{W}_* \in \mathbb{D}$ denote the true DAG matrix. Assume the data $x$ is generated by the linear SEM with independent and Gaussian noise $n$*

$$x = (I - \mathbf{W}_*^\top)^{-1}n, \qquad \mathbb{E}[n] = 0, \qquad \mathbb{E}[nn^\top] = \sigma^2 I_d. \tag{14}$$

*Consider the objective in (9):*

$$\min_{\mathbf{W} \in \mathbb{D}} L(\mathbf{W}) = \min_{\mathbf{W} \in \mathbb{D}} \mathbb{E}_{x \sim \mathcal{D}} \left[ l(\mathbf{W}; x) \right], \tag{15}$$

*where $l(\mathbf{W}; x) \overset{def}{=} \frac{1}{2}\|x - \mathbf{W}^\top x\|^2$.*

*Then $\mathbf{W}_*$ is a global minimizer of $L(\mathbf{W})$ over $\mathbb{D}$:*

$$\mathbf{W}_* = \arg\min_{\mathbf{W} \in \mathbb{D}} L(\mathbf{W}).$$

*Proof.* For any $\mathbf{W} \in \mathbb{D}$ and any $x$,

$$l(\mathbf{W}; x) = \tfrac{1}{2}\|x - \mathbf{W}^\top x\|^2 = \tfrac{1}{2}\|(I - \mathbf{W}^\top)x\|^2.$$

Using the SEM (14), we substitute $x = (I - \mathbf{W}_*^\top)^{-1}n$:

$$x - \mathbf{W}^\top x = (I - \mathbf{W}^\top)(I - \mathbf{W}_*^\top)^{-1}n.$$

Define

$$M(\mathbf{W}) := (I - \mathbf{W}^\top)(I - \mathbf{W}_*^\top)^{-1}.$$

Then

$$\|x - \mathbf{W}^\top x\|^2 = \|M(\mathbf{W})n\|^2 = n^\top M(\mathbf{W})^\top M(\mathbf{W})\, n.$$

Taking expectation and using $\mathbb{E}[nn^\top] = \sigma^2 I_d$,

$$\begin{aligned}
L(\mathbf{W}) &= \tfrac{1}{2}\, \mathbb{E}\big[\|x - \mathbf{W}^\top x\|^2\big] \\
&= \tfrac{1}{2}\, \mathbb{E}\big[n^\top M(\mathbf{W})^\top M(\mathbf{W})\, n\big] \\
&= \tfrac{1}{2}\, \mathrm{Tr}\big(M(\mathbf{W})^\top M(\mathbf{W})\, \mathbb{E}[nn^\top]\big) \\
&= \tfrac{\sigma^2}{2}\, \mathrm{Tr}\big(M(\mathbf{W})^\top M(\mathbf{W})\big) \\
&= \tfrac{\sigma^2}{2}\, \|M(\mathbf{W})\|_F^2.
\end{aligned}$$

Let us compute the value of the function at $\mathbf{W}_*$:

$$L(\mathbf{W}_*) = \tfrac{\sigma^2}{2}\|M(\mathbf{W}_*)\|_F = \tfrac{\sigma^2}{2}\|I_d\|_F = \tfrac{\sigma^2 d}{2}$$

Because $\mathbf{W}, \mathbf{W}_* \in \mathbb{D}$ are both DAGs and hence nilpotent matrices. In particular,

$$\det(I - \mathbf{W}) = 1 \quad \text{for all } \mathbf{W} \in \mathbb{D}, \tag{16}$$

hence

$$\det M(\mathbf{W}) = 1 \quad \text{for all } \mathbf{W} \in \mathbb{D}, \tag{17}$$

Let $s_1(\mathbf{W}), \ldots, s_d(\mathbf{W}) > 0$ be the singular values of $M(\mathbf{W})$. Then

$$\|M(\mathbf{W})\|_F^2 = \sum_{i=1}^d s_i(\mathbf{W})^2, \qquad \prod_{i=1}^d s_i(\mathbf{W}) = |\det M(\mathbf{W})| = 1,$$

where the second equality is by (17).

Apply the arithmetical mean and geometrical mean inequality to the nonnegative numbers $s_i(\mathbf{W})^2$:

$$\frac{1}{d} \sum_{i=1}^d s_i(\mathbf{W})^2 \geq \left(\prod_{i=1}^d s_i(\mathbf{W})^2\right)^{1/d} = \left(\prod_{i=1}^d s_i(\mathbf{W})\right)^{2/d} = 1.$$

Thus

$$\|M(\mathbf{W})\|_F^2 = \sum_{i=1}^d s_i(\mathbf{W})^2 \geq d. \tag{18}$$

Equality in (18) holds if and only if all $s_i(\mathbf{W})^2$ are equal, i.e. $s_1(\mathbf{W}) = \cdots = s_d(\mathbf{W}) = 1$.

Finally, by (18),

$$L(\mathbf{W}) = \frac{\sigma^2}{2} \|M(\mathbf{W})\|_F^2 \geq \frac{\sigma^2}{2} d,$$

with equality if and only if $M(\mathbf{W}) = I$. The condition $M(\mathbf{W}) = I$ is equivalent to

$$(I - \mathbf{W})(I - \mathbf{W}_*)^{-1} = I \iff I - \mathbf{W} = I - \mathbf{W}_* \iff \mathbf{W} = \mathbf{W}_*.$$

Therefore, $\mathbf{W}_*$ is the unique global minimizer of $L(\mathbf{W})$ over $\mathbb{D}$. $\qquad \square$

Now, we move to the proposed method.

**Theorem 8.** *For an $L_1$-smooth function $F(\mathbf{W}) = \mathbb{E}_{x \sim \mathcal{D}}[l(\mathbf{W}; x)]$ restricted in a domain of radius $R$, $\|x - y\| \leq R$, $\forall x, y \in$ dom $F$, consider $\mathcal{A}_2$ in the Algorithm 1 be chosen as Universal Stochastic Gradient Method (Rodomanov et al., 2024). Running $\mathcal{A}_2$ for $T$ SGD-type steps accessing $\sigma_1$-stochastic gradients (Theorem 1) in the $\pi$-subspace $\mathbb{D}_\pi$ converges to a minimum of problem (9) with additional subspace constraints at the rate*

$$\mathbb{E}\left[F(\mathbf{W}_T) - \underset{\substack{\mathbf{W} \in \mathbb{D}_\pi, \\ ord(\mathbf{W})=\pi}}{\arg\min} F(\mathbf{W})\right] \leq \mathcal{O}\left(\frac{\sigma_1 R}{\sqrt{T}} + \frac{L_1 R^2}{T}\right).$$

*Proof.* A direct consequence of the convergence guarantees of the Universal Stochastic Gradient Method, Theorem 4.2 of Rodomanov et al. (2024). $\qquad \square$

**Theorem 9.** *For an $L_1$-smooth function $F(\mathbf{W}) = \mathbb{E}_{x \sim \mathcal{D}}[l(\mathbf{W}; x)]$ restricted in a domain of radius $R$, $\|x - y\| \leq R$, $\forall x, y \in$ dom $F$, Algorithm 2 with Universal Stochastic Gradient Method (Rodomanov et al., 2024) as $\mathcal{A}_1$ and $\mathcal{A}_2$ with converges to a local minimum of problem (9).*

*Proof.* We denote a subspace minimum of problem (9) as $\mathbf{W}_{\pi_k}^* = \arg\min_{\substack{\mathbf{W} \in \mathbb{D}_\pi, \\ ord(\mathbf{W})=\pi}} F(\mathbf{W})$. There are two cases. The first case, $\mathbf{W}_{\pi_k^*}$ is a local minimum of a general problem (9). Then, by Theorem 8, the method converges to the local minimum. The second case, the subspace minimum $\mathbf{W}_{\pi_k}^*$ is not a local minimum as there exists an orthogonal subspace $\mathbb{D}_{\tilde{\pi}} \perp \mathbb{D}_\pi$ such that $\mathbf{W}_{\pi_k}^* \in \mathbb{D}_{\tilde{\pi}}$ and $F(\mathbf{W}_{\pi_k}^*) > F(\mathbf{W}_{\tilde{\pi}}^*)$. By steps from $\mathcal{A}_1^{k+1}$, the method decreases towards $F(\mathbf{W}_{\tilde{\pi}}^*)$. To fully guarantee the convergences, one can memorize the visited subspaces and forbid projecting on them. Then, $\mathbb{D}_{\pi_{k+1}}$ is not visited subspace. $\qquad \square$

## C  DETAILED EXPERIMENT DESCRIPTION

**Computing.** Our experiments were carried out on a machine equipped with 80 CPUs and one NVIDIA Quadro RTX A6000 48GB GPU. Each experiment was allotted a maximum wall time of 36 hours as in DAGMA Bello et al. (2022).

**Graph Models.** In our experimental simulations, we generate graphs using two established random graph models:

- **Erdős-Rényi (ER) graphs:** These graphs are constructed by independently adding edges between nodes with a uniform probability. We denote these graphs as $\text{ER}_k$, where $kd$ represents the expected number of edges.

- **Scale-Free (SF) graphs**: These graphs follow the preferential attachment process as described in Barabási & Albert (1999). We use the notation $\text{SF}_k$ to indicate a scale-free graph with expected $kd$ edges and an attachment exponent of $\beta = 1$, consistent with the preferential attachment process. Since we focus on directed graphs, this model corresponds to Price's model, a traditional framework used to model the growth of citation networks.

It is important to note that ER graphs are inherently undirected. To transform them into Directed Acyclic Graphs (DAGs), we generate a random permutation of the vertex labels from 1 to $d$, then orient the edges according to this ordering. For SF graphs, edges are directed as new nodes are added, ensuring that the resulting graph is a DAG. After generating the ground-truth DAG, we simulate the structural equation model (SEM) for linear cases, conducting experiments accordingly.

**Metrics.** The performance of each algorithm is assessed using the following four key metrics:

- **Structural Hamming Distance (SHD):** A widely used metric in structure learning that quantifies the number of edge modifications (additions, deletions, and reversals) required to transform the estimated graph into the true graph.

- **True Positive Rate (TPR):** This metric calculates the proportion of correctly identified edges relative to the total number of edges in the ground-truth DAG. It is also known as **recall**.

- **Precision:** is the proportion of all the model's positive classifications that are actually positive.

- **F1 Score:** is the harmonic mean of precision and recall.

- **False Positive Rate (FPR):** This measures the proportion of incorrectly identified edges relative to the total number of absent edges in the ground-truth DAG.

- **Runtime:** The time taken by each algorithm to complete its execution provides a direct measure of the algorithm's computational efficiency.

- **Stochastic gradient computations:** Number of gradient computed.

**Linear SEM.** In the linear case, the functions are directly parameterized by the weighted adjacency matrix $W$. Specifically, the system of equations is given by $X_i = \mathbf{X}\mathbf{W}_i + N_i$, where $\mathbf{W} = [\mathbf{W}_1|\cdots|\mathbf{W}_d] \in \mathbb{R}^{d \times d}$, and $N_i \in \mathbb{R}$ represents the noise. The matrix $\mathbf{W}$ encodes the graphical structure, meaning there is an edge $X_j \to X_i$ if and only if $W_{j,i} \neq 0$. Starting with a ground-truth DAG $B \in \{0,1\}^{d \times d}$ obtained from one of the two graph models, either ER or SF, edge weights were sampled independently from $\text{Unif}[-2, -0.5] \cup [0.5, 2]$ to produce a weight matrix $\mathbf{W} \in \mathbb{R}^{d \times d}$. Using this matrix $\mathbf{W}$, the data $X = X\mathbf{W} + N$ was sampled under the following three noise models:

- **Gaussian noise:** $N_i \sim N(0,1)$ for all $i \in [d]$,

- **Exponential noise:** $N_i \sim \text{Exp}(1)$ for all $i \in [d]$,

- **Gumbel noise:** $N_i \sim \text{Gumbel}(0,1)$ for all $i \in [d]$.

Using these noise models, random datasets $X \in \mathbb{R}^{n \times d}$ were generated by independently sampling the rows according to one of the models described above. Unless otherwise specified, we generate the same number of samples $n \in \{5000, 10000\}$ for training and validation datasets, respectively.

The implementation details of the baseline methods are as follows:

- FGES **(Meek, 1997; Chickering, 2002)** using the FGES algorithm found in the py-tetrad package Scheines et al. (1998) in `https://github.com/cmu-phil/py-tetrad`.

- PC (**Spirtes & Glymour, 1991; Ramsey et al., 2012**) using the PC algorithm from the py-tetrad package Scheines et al. (1998) in `https://github.com/cmu-phil/py-tetrad`.

- NOTEARS (**Zheng et al., 2018**) using the authors' publicly available Python code, which can be found at `https://github.com/xunzheng/notears`. This method employs a least squares score function, and we used their default set of hyperparameters without modification. We used the default choice of $\lambda = 0.1$ as in authors' code.

- GOLEM (**Ng et al., 2020**) using the authors' Python code, available at `https://github.com/ignavierng/golem`, along with their PyTorch version at `https://github.com/huawei-noah/trustworthyAI/blob/master/gcastle/castle/algorithms/gradient/notears/torch/golem_utils/golem_model.py`. We adopted the default hyperparameter settings, specifically $\lambda_1 = 0.02$ and $\lambda_2 = 5$. Additional details of GOLEM are listed in Ng et al. (2020)(Appendix F).

- NOCURL (**Yu et al., 2021**) using the authors' publicly available code at `https://github.com/fishmoon1234/DAG-NoCurl`. We used the default choice of hyperparameters.

- DAGMA (**Bello et al., 2022**) using the authors' Python code, which is available at `https://github.com/kevinsbello/dagma`. We used the default choice of hyperparameters.

**Thresholding.** Following the approach taken in previous studies, including the baseline methods (Zheng et al., 2018; Ng et al., 2020; Yu et al., 2021; Bello et al., 2022), for all the methods, we apply a final thresholding step of 0.3 to effectively reduce the number of false discoveries.

# D  SUPPLEMENTARY EXPERIMENTS RESULTS

## D.1  STRUCTURE RECOVERY PERFORMANCE

In addition to the results presented in the main paper (Section 5.2), we include extended evaluations on ER6 graphs under Gaussian noise with non-equal variances (NV). As illustrated in Figure 6, $\psi$DAG consistently achieves the best performance in most metrics, including the Structural Hamming distance (SHD), precision, and F1 score, for all noise levels and graph sizes. In particular, while $\psi$DAG slightly trails FGES in recall (or TPR) for graphs with $d > 50$, FGES exhibits a significantly worse SHD, precision, and F1 score in those regimes, suggesting that it produces denser graphs with more false positives. This reinforces that $\psi$DAG not only maintains structural accuracy, but also avoids overfitting, particularly in complex, high-noise environments. These results highlight the robustness of our approach across graph densities and noise heterogeneity.

We present a comparative analysis of structure recovery and runtime performance under Gaussian noise with equal variances across two random graph types: ER2 and ER3. Figure 7 illustrates Structural Hamming Distance (SHD) and runtime (in seconds) across varying node sizes $d \in \{25, 50, 100, 500\}$. Compared methods include constraint-based (PC), score-based (FGES), and gradient-based approaches (NOTEARS, GOLEM, NOCURL, DAGMA, and $\psi$DAG). As the number of nodes increases, PC and FGES exhibit a marked rise in SHD, indicating limited scalability in structural accuracy. NOTEARS and GOLEM, while competitive on small scales, incur significantly higher runtime as $d$ grows, making them less practical for large graphs. In contrast, $\psi$DAG maintains a low SHD and a consistently competitive runtime, demonstrating both scalability and robustness in high-dimensional settings.

To evaluate structure recovery under more challenging conditions, we include additional results on large-scale ER2 and ER3 graphs ($d \in \{400, 500, 800, 1000\}$) with Gaussian noise exhibiting non-equal variances (NV) across varying noise ratios. As shown in Figure 8, $\psi$DAG consistently achieves the best or near-best performance across most metrics, demonstrating robustness even at a large scale.

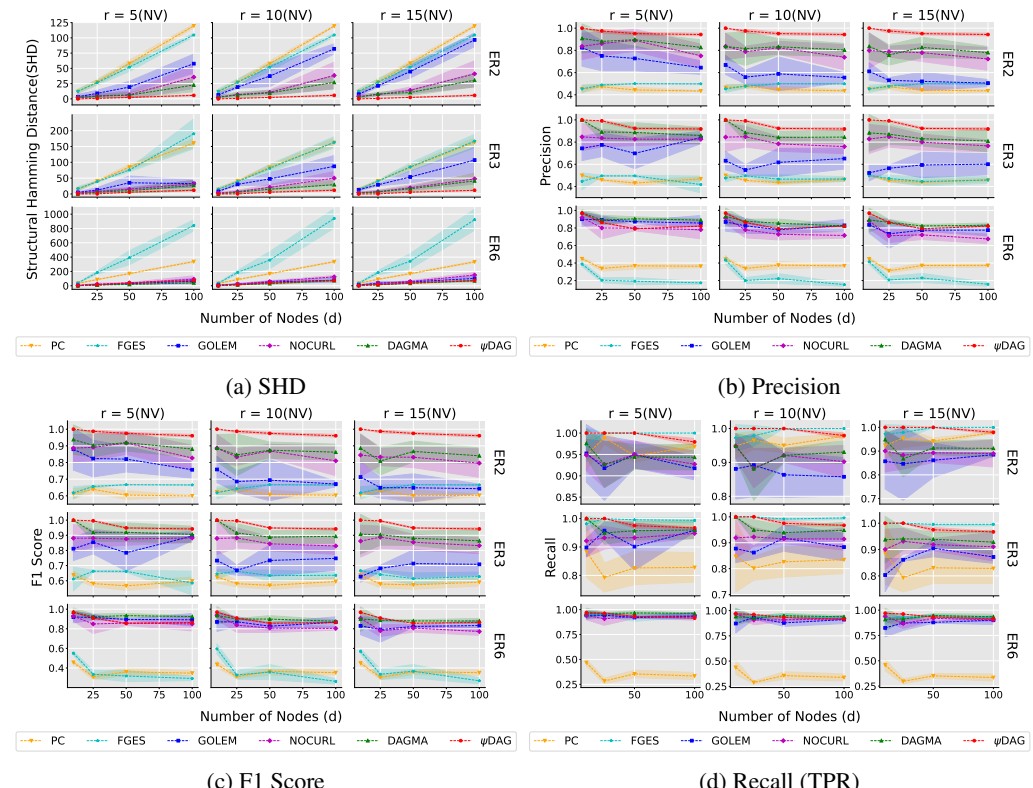

(a) SHD           (b) Precision

(c) F1 Score           (d) Recall (TPR)

Figure 6: Structure recovery performance under Gaussian noise with non-equal variances (NV) across varying noise ratios ($r \in \{5, 10, 15\}$). Rows correspond to different random graph types, and columns represent increasing noise ratios. Metrics reported include (from left to right): Structural Hamming Distance (SHD, lower is better), Precision, F1 score, and Recall (or TPR) (all higher is better). Each method's mean performance is shown, with standard error indicated by shaded regions around the curves.

## D.2 SCALABILITY COMPARISON

We provide complementary scalability results to those reported in the main paper (Section 5.3). $\psi$DAG scales efficiently to graphs with up to $d = 10{,}000$ nodes, as shown in Figure 9. In sparse settings such as ER2 and SF2, it converges within a few hours even for the largest graphs. In contrast, the runtime of GOLEM, NOCURL, and DAGMA increases sharply with graph size. On ER2 graphs, both GOLEM and NOCURL exceed the 36-hour runtime limit for $d \geq 3000$, while DAGMA fails to complete within the time budget for $d \geq 5000$.

We also observe frequent convergence failures among baselines. For instance, NOCURL fails to converge on SF2 graphs with Gaussian-EV noise when $d = 500$, and entirely fails on ER6. Both GOLEM and DAGMA exhibit non-convergence in at least one of the three random seeds on ER6. All such failed runs are excluded from reported statistics. In contrast, $\psi$DAG converges in all runs and maintains competitive runtime performance even at large scales, highlighting both its robustness and practical efficiency.

We present results across combinations of the number of vertices $d \in \{10, 50, 100, 500, 1000, 3000, 5000, 10000\}$, graph types $\in \{ER, SF\}$, graph densities $k \in \{2, 4, 6\}$, and noise types (Gaussian, Exponential, and Gumbel). Figures are grouped by noise type and graph size, with subplots showing results for $\psi$DAG, GOLEM, and DAGMA.

**ER graph types:** Figures 10 and 11 report results on ER2 graphs; Figures 12 and 13 on ER4 graphs; and Figure 14 on ER6 graphs.

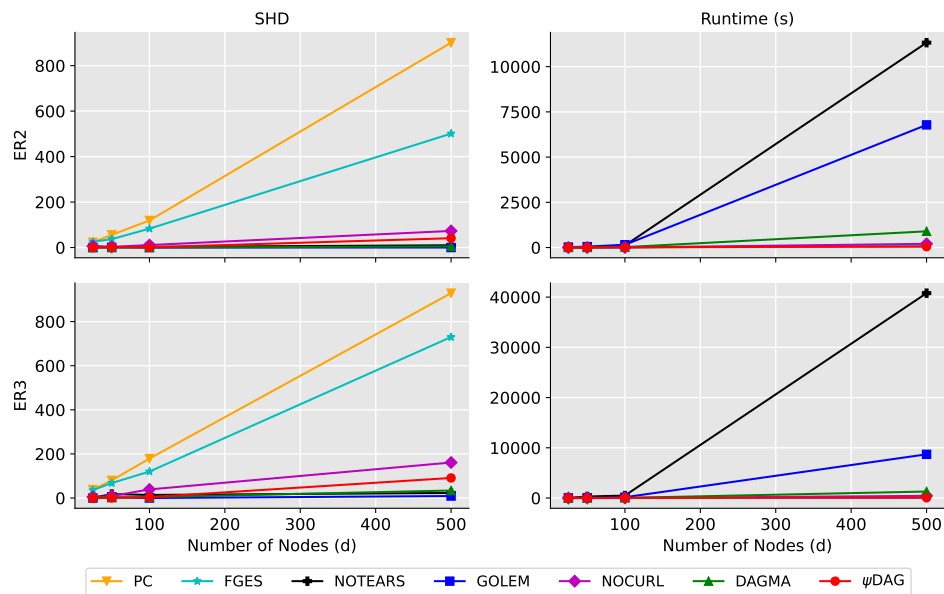

Figure 7: Comparison of Structural Hamming Distance (SHD ↓) and Runtime (in seconds ↓) across constraint-based (PC), score-based (FGES), and gradient-based approaches (NOTEARS, GOLEM, NOCURL, DAGMA, and $\psi$DAG) on ER2 and ER3 Gaussian-EV random graphs. Lower SHD indicates better structural accuracy; lower runtime indicates greater efficiency.

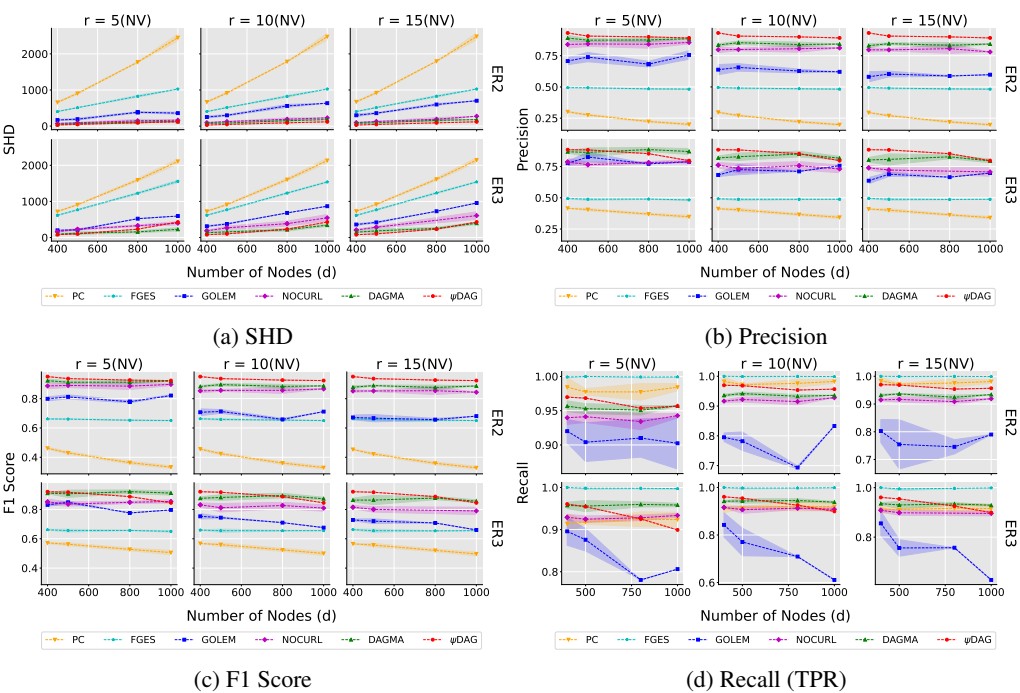

(a) SHD

(b) Precision

(c) F1 Score

(d) Recall (TPR)

Figure 8: Structure recovery performance under Gaussian noise with non-equal variances (NV) across varying noise ratios ($r \in \{5, 10, 15\}$). Rows correspond to ER2 and ER3 graphs with $d \in \{400, 500, 800, 1000\}$, and columns represent increasing noise ratios. Metrics reported include (from left to right): Structural Hamming Distance (SHD, lower is better), Precision, F1 score, and Recall (or TPR) (all higher is better). Each method's mean performance is shown, with standard error indicated by shaded regions around the curves.

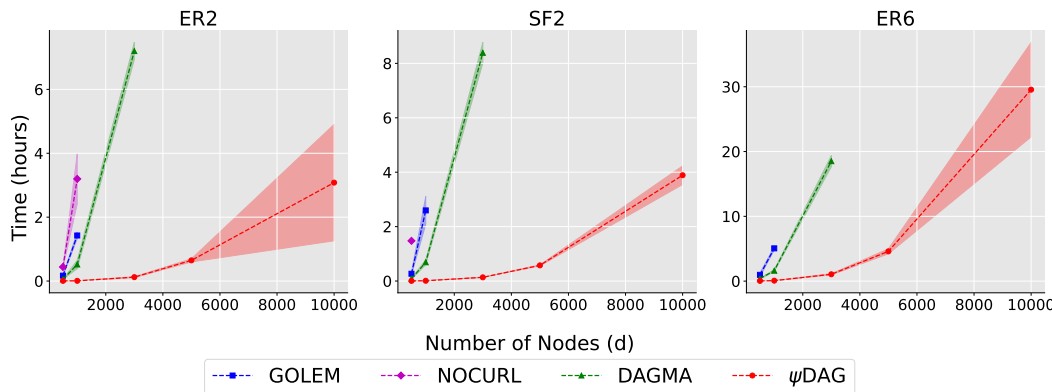

Figure 9: Runtime comparison of $\psi$DAG, GOLEM, NOCURL, and DAGMA on ER2, SF2 and ER6 graphs with $d \in \{500, 1000, 3000, 5000, 10000\}$. The noise distribution is Gaussian with equal variances. $\psi$DAG scales efficiently to 10,000 nodes, while the baselines exhibit sharp runtime increases or fail to complete within the 36-hour time budget. Notably, GOLEM, NOCURL, and DAGMA fail to converge or exceed the time limit in multiple settings, especially for ER6. All non-converging runs were excluded from the figures.

**SF graph types:** Figures 15 and 16 report results on SF2 graphs; Figures 17 and 18 on SF4 graphs; and Figure 19 on SF6 graphs.

We report the decrease in functional value over both **(i)** elapsed time and **(ii)** number of gradient evaluations, the latter serving as a proxy for computational effort.

Figure 11b highlights that DAGMA requires substantially more gradient computations compared to both $\psi$DAG and GOLEM, further emphasizing the efficiency of our approach.

### D.3 SMALL TO MODERATE NUMBER OF NODES

Our experiments demonstrate that while number of nodes is small, $d < 100$, GOLEM is more stable than DAGMA, and $\psi$DAG method is the most stable. While DAGMA shows impressive speed for smaller node sets, the number of iterations required is still higher than both GOLEM and our method. Across all scenarios, $\psi$DAG consistently demonstrates faster convergence compared to the other approaches, requiring fewer iterations to reach the desired solution.

### D.4 LARGE NUMBER OF NODES

For graphs with a large number of nodes $d \in \{5000, 10000\}$, we were unable to run neither of the baselines, and consequently, Figure 20 includes only one algorithm. GOLEM was not feasible due to its computation time exceeding 350 hours. DAGMA was impossible as its runs led to kernel crashes. In all cases, we utilized a training set of 5,000 samples and a validation set of 10,000 samples.

### D.5 DENSER GRAPHS

For a thorough comparison, in Figures 14 and 19, we compare graph structures ER6 and SF6 under the Gaussian noise type. Plots indicate that while DAGMA exhibits a fast runtime when the number of nodes is small, $d < 100$, it requires more iterations to achieve convergence. Algorithm $\psi$DAG consistently outperforms GOLEM and DAGMA in both training time and a number of stochastic gradient computations, and the difference is more pronounced for a larger number of nodes and denser graphs.

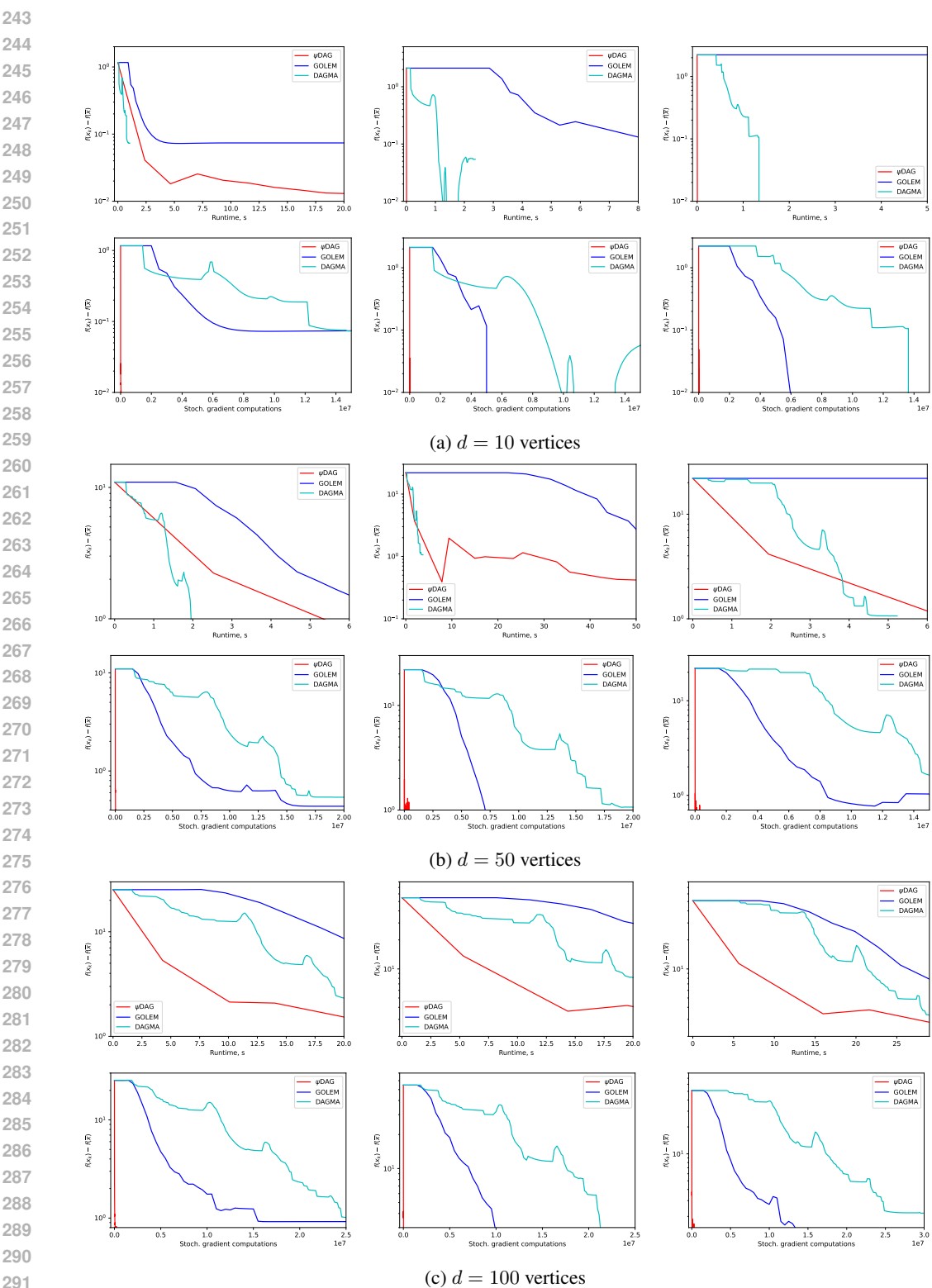

(a) $d = 10$ vertices

(b) $d = 50$ vertices

(c) $d = 100$ vertices

Figure 10: Linear SEM methods on graphs of type ER2 with different noise distributions: Gaussian (first), exponential (second), Gumbel (third).

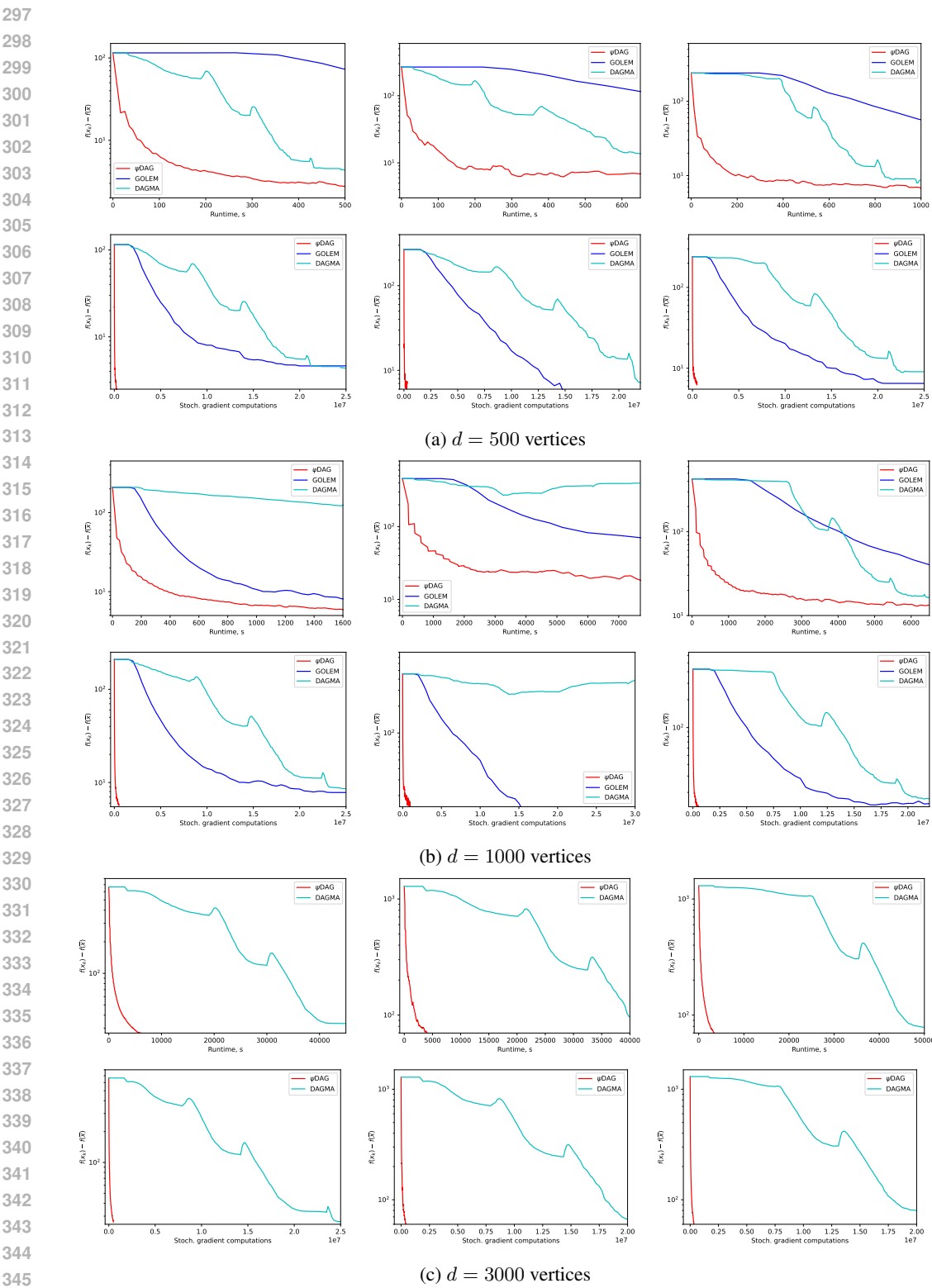

(a) $d = 500$ vertices

(b) $d = 1000$ vertices

(c) $d = 3000$ vertices

Figure 11: Linear SEM methods on graphs of type ER2 with different noise distributions: Gaussian (first), exponential (second), Gumbel (third).

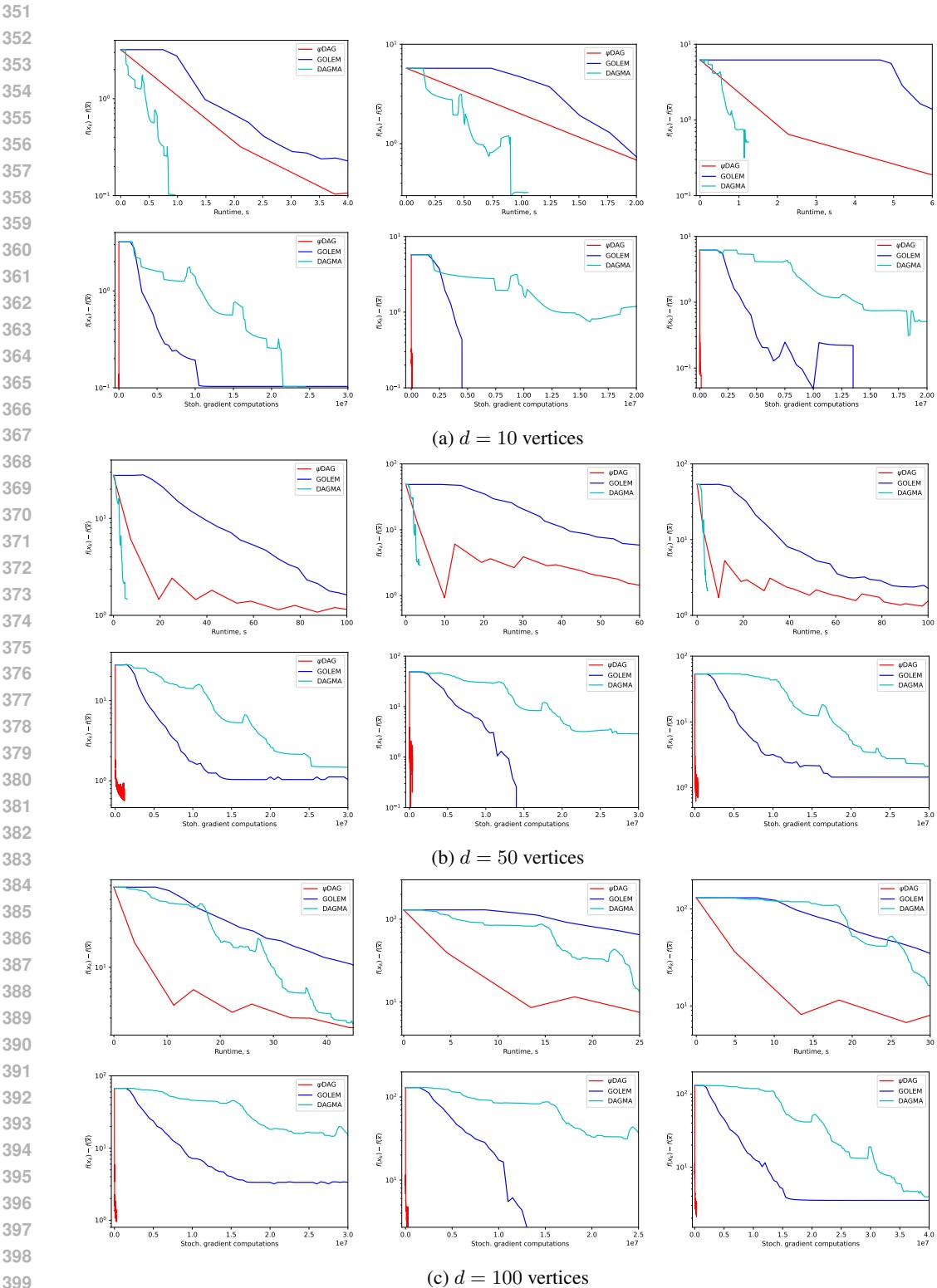

(a) $d = 10$ vertices

(b) $d = 50$ vertices

(c) $d = 100$ vertices

Figure 12: Linear SEM methods on graphs of type ER4 with different noise distributions: Gaussian (first), exponential (second), Gumbel (third).

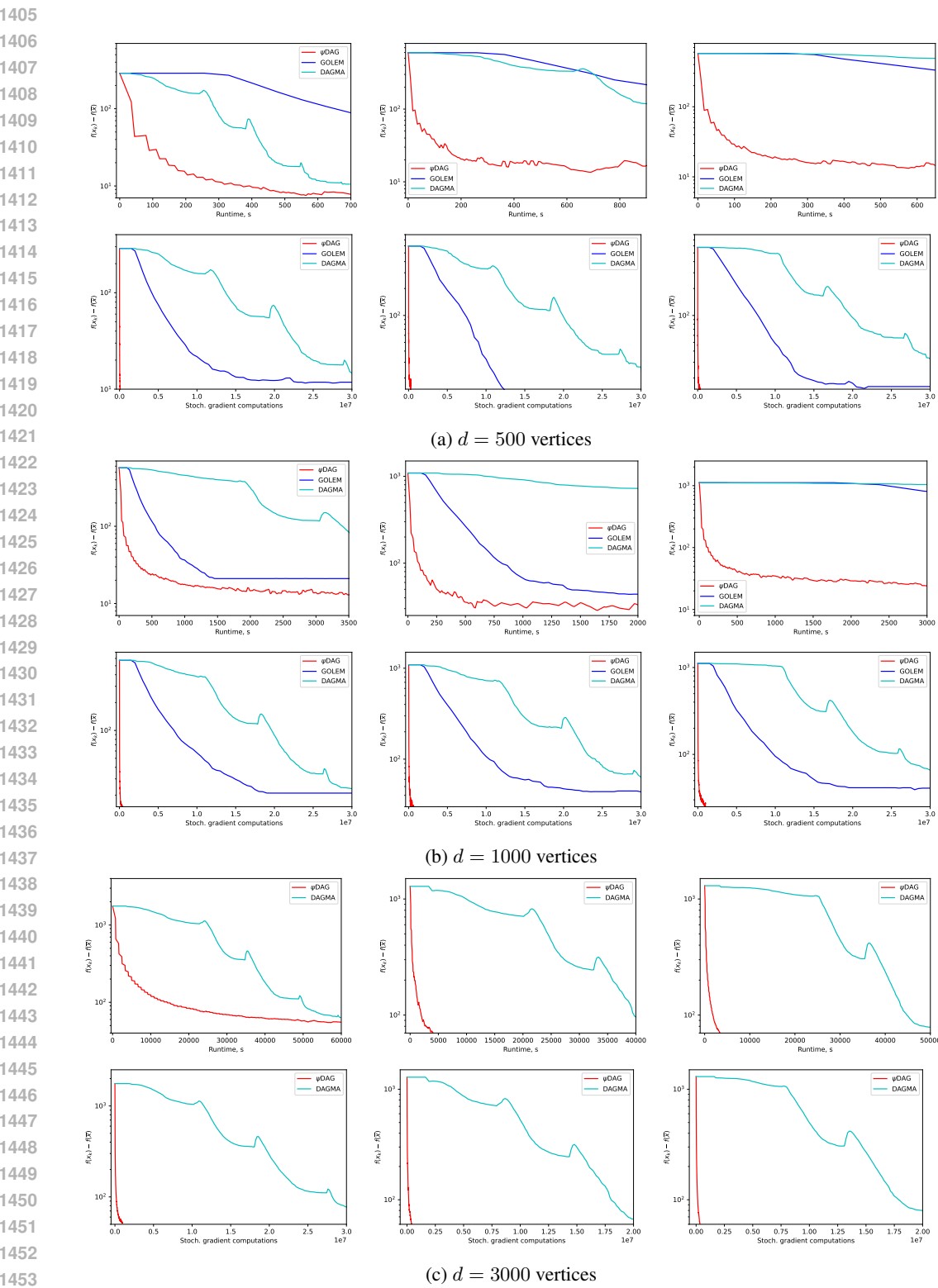

(a) $d = 500$ vertices

(b) $d = 1000$ vertices

(c) $d = 3000$ vertices

Figure 13: Linear SEM methods on graphs of type ER4 with different noise distributions: Gaussian (first), exponential (second), Gumbel (third).

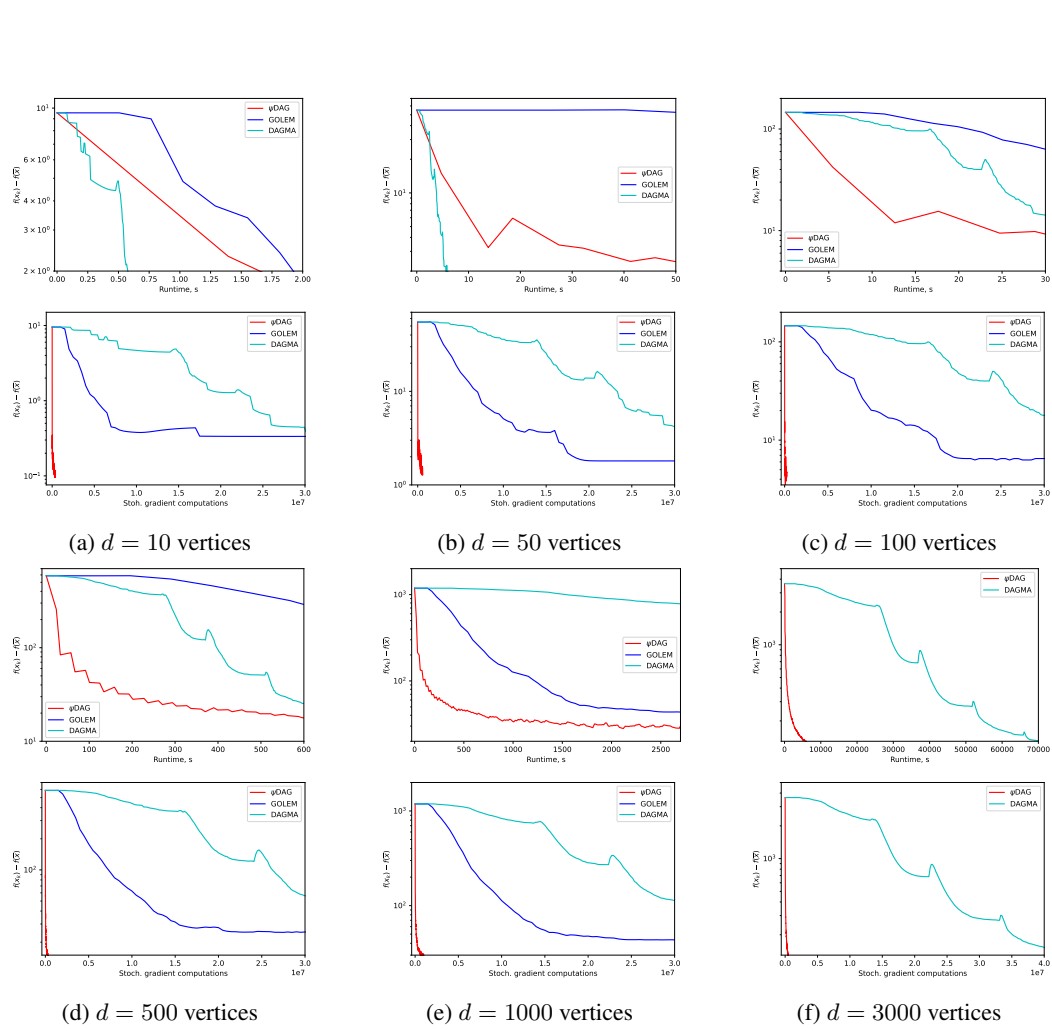

(a) $d = 10$ vertices

(b) $d = 50$ vertices

(c) $d = 100$ vertices

(d) $d = 500$ vertices

(e) $d = 1000$ vertices

(f) $d = 3000$ vertices

Figure 14: Linear SEM methods on graphs of type ER6 with the Gaussian noise distribution.

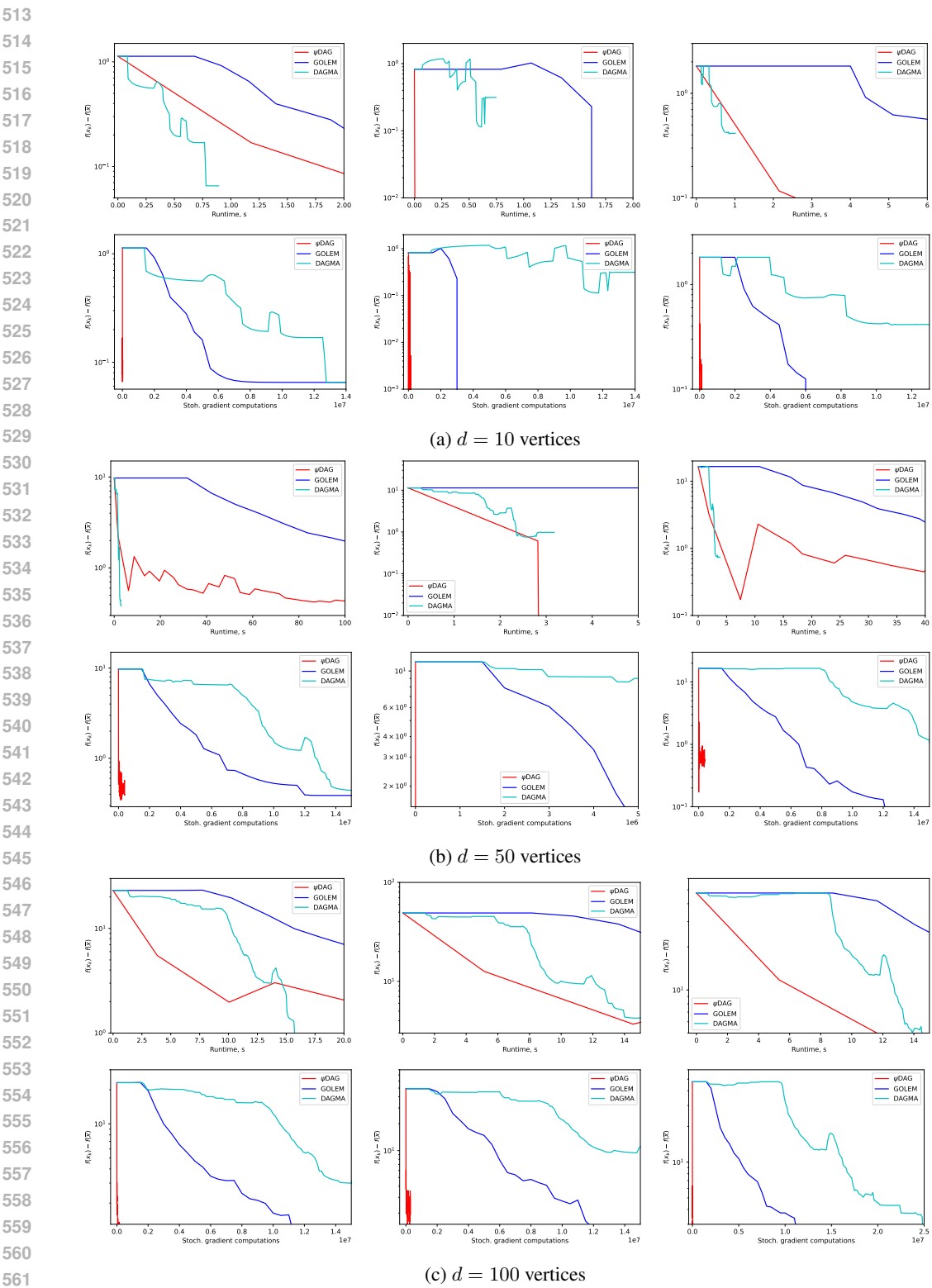

(a) $d = 10$ vertices

(b) $d = 50$ vertices

(c) $d = 100$ vertices

Figure 15: Linear SEM methods on graphs of type SF2 with different noise distributions: Gaussian (first), exponential (second), Gumbel (third).

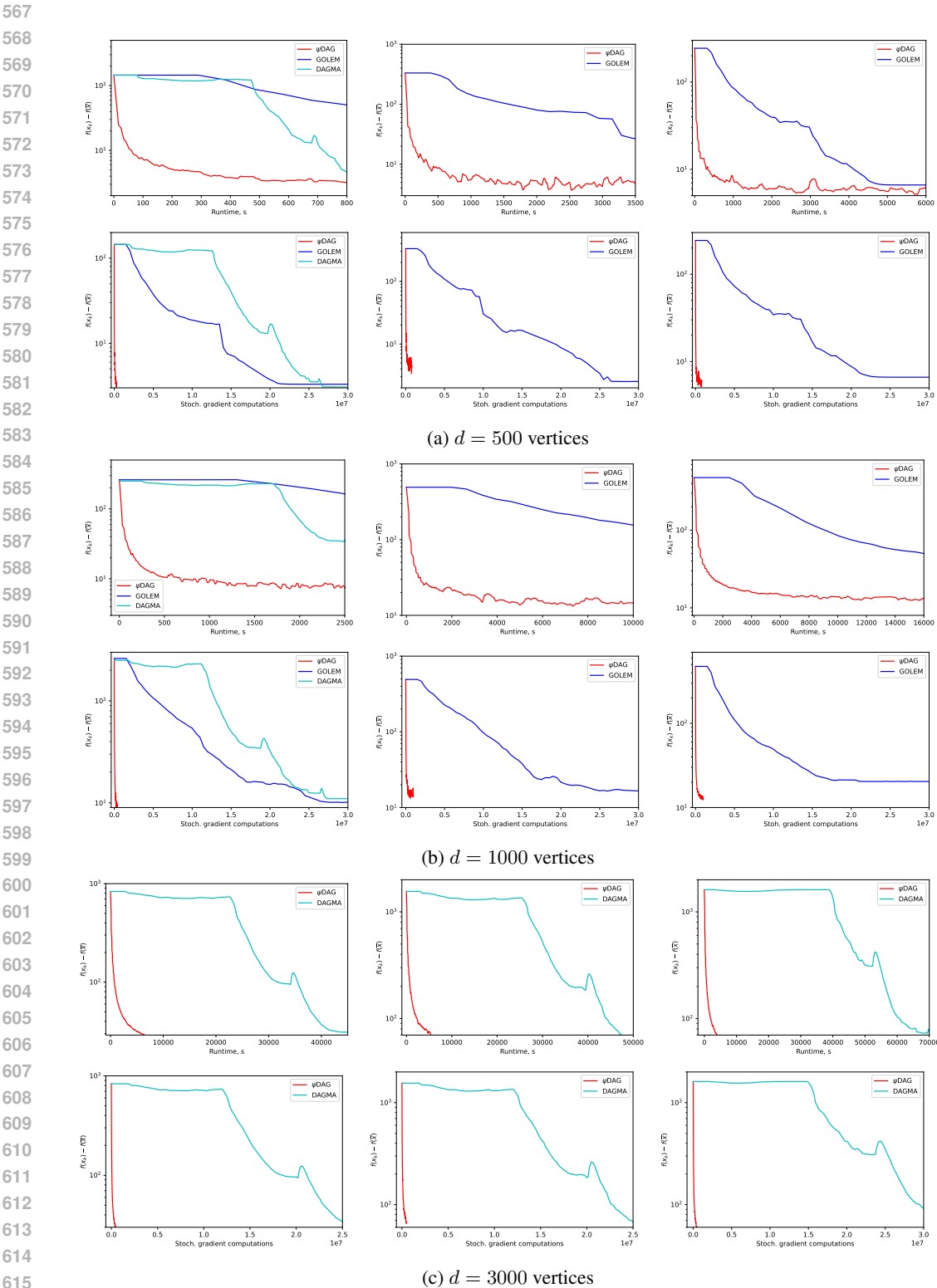

(a) $d = 500$ vertices

(b) $d = 1000$ vertices

(c) $d = 3000$ vertices

Figure 16: Linear SEM methods on graphs of type SF2 with different noise distributions: Gaussian (first), exponential (second), Gumbel (third).

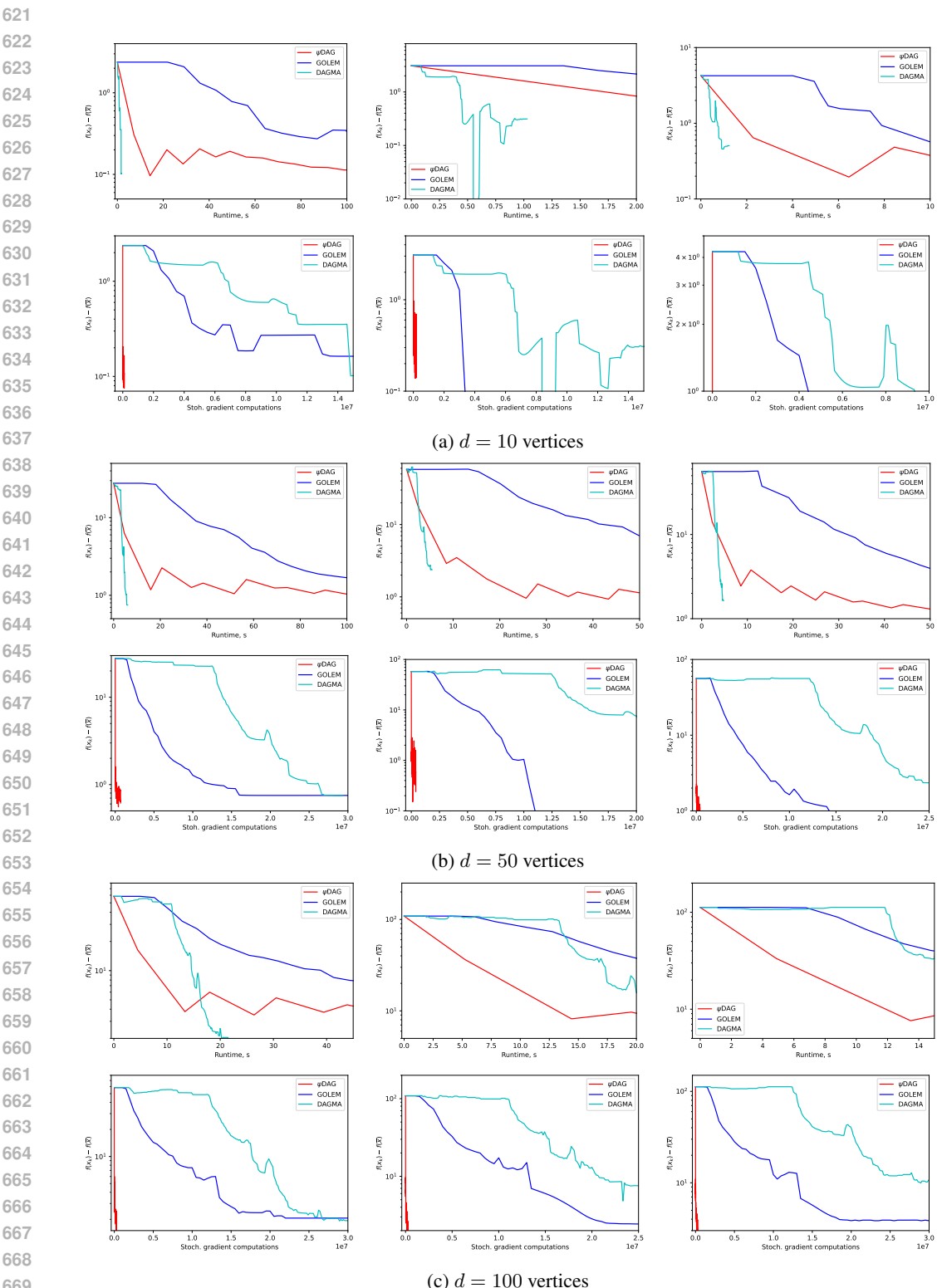

(a) $d = 10$ vertices

(b) $d = 50$ vertices

(c) $d = 100$ vertices

Figure 17: Linear SEM methods on graphs of type SF4 with different noise distributions: Gaussian (first), exponential (second), Gumbel (third).

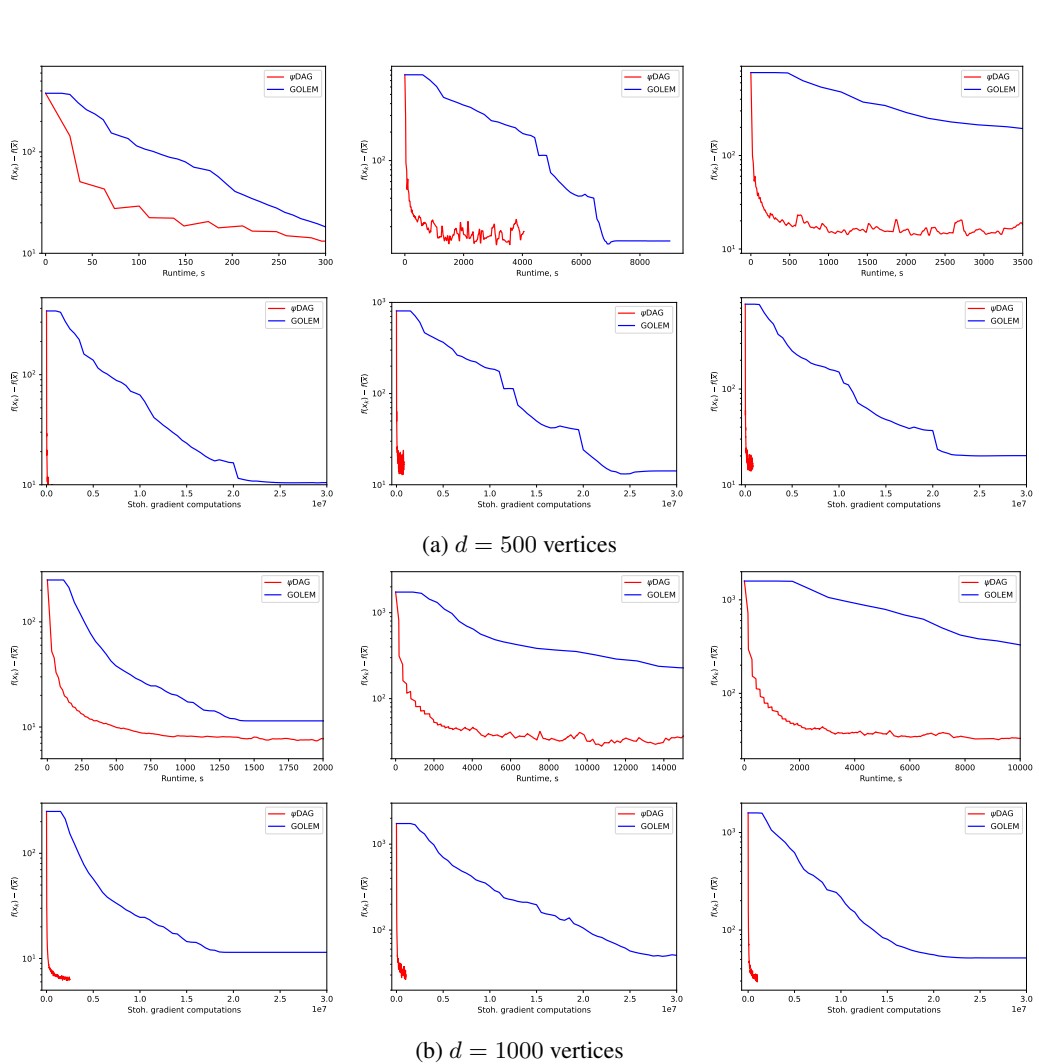

(a) $d = 500$ vertices

(b) $d = 1000$ vertices

Figure 18: Linear SEM methods on SF4 graphs with different noise distributions: Gaussian (first), exponential (second), Gumbel (third).

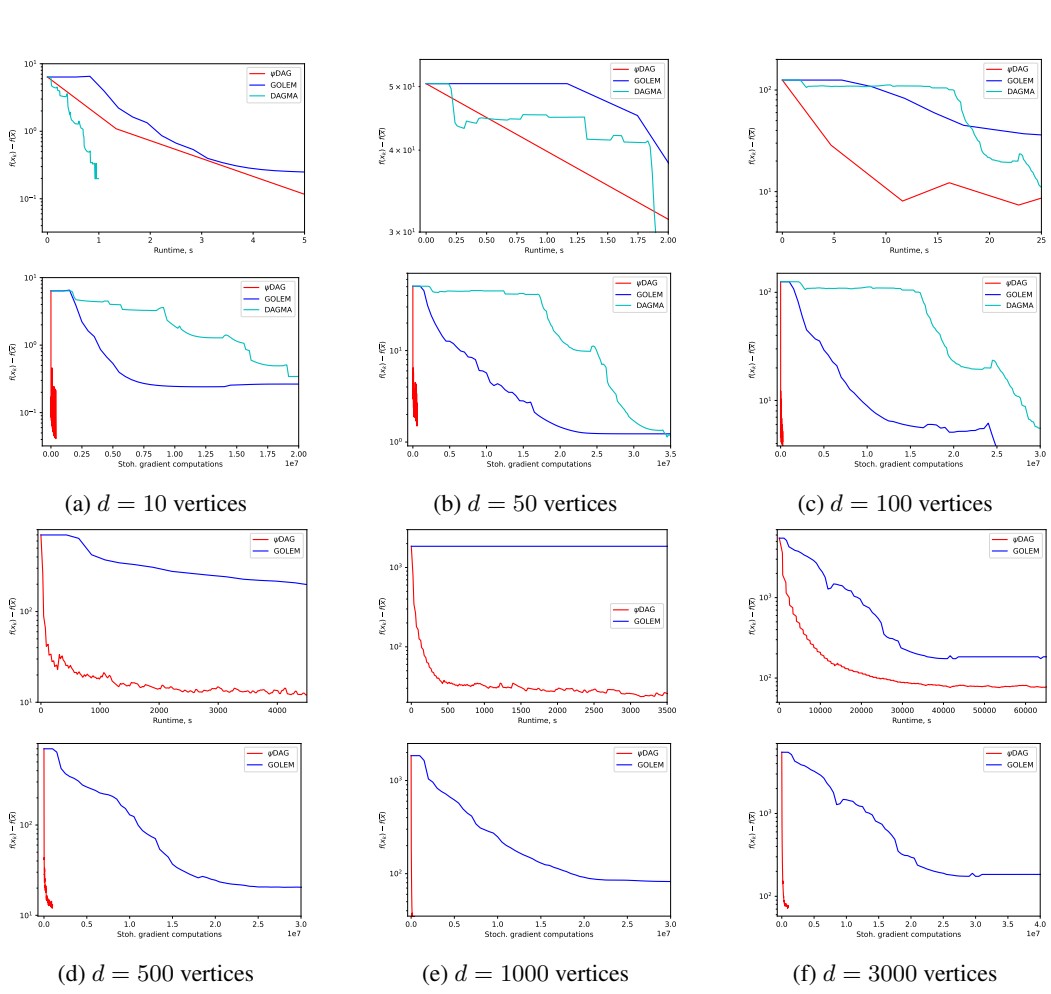

Figure 19: Linear SEM methods on graphs of type SF6 with the Gaussian noise distribution.

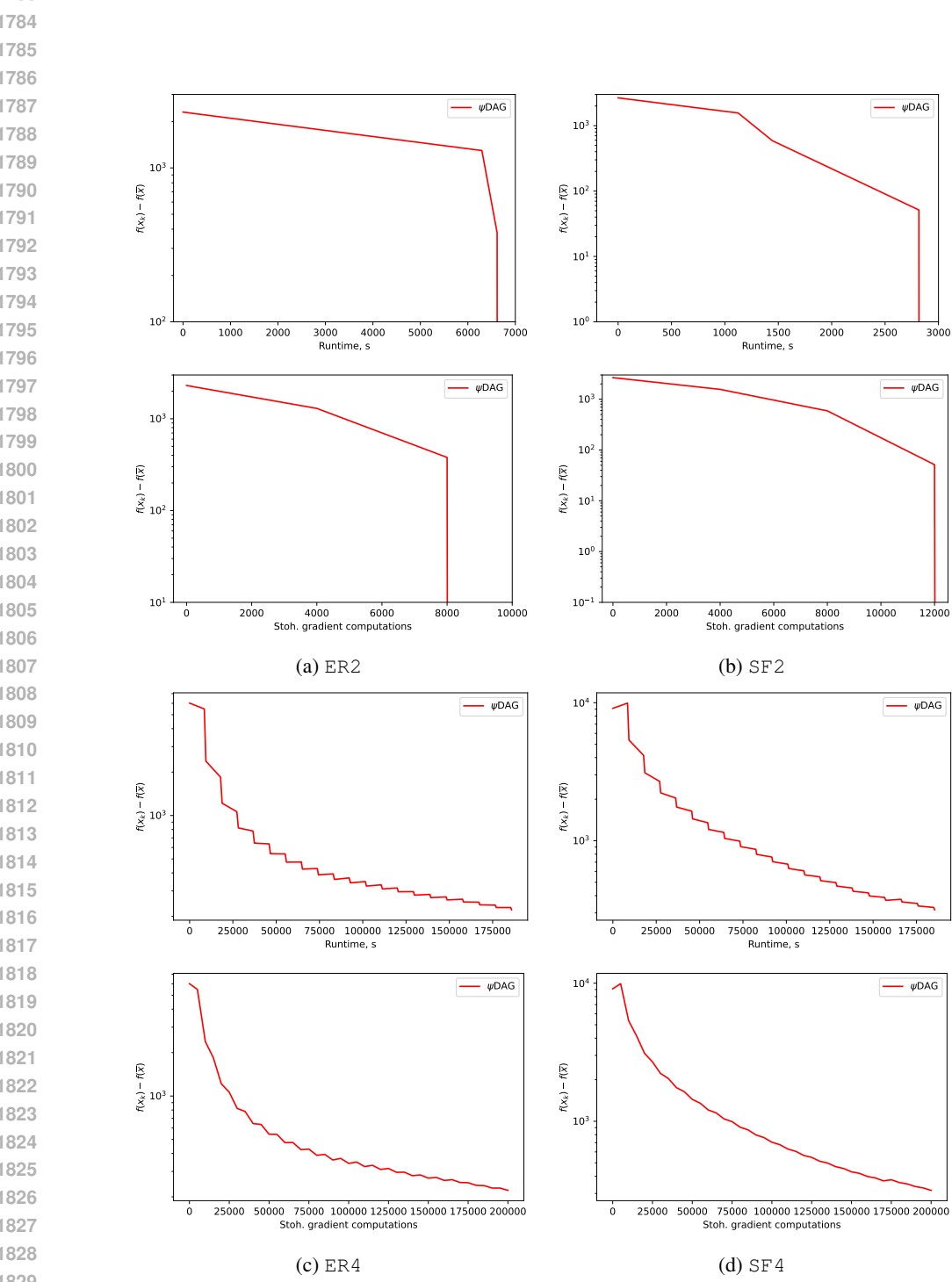

Figure 20: $\psi$DAG method for graph types ER2, ER4, SF2 and SF4 graphs with $d = 10000$ and Gaussian noise. Other linear SEM methods do not converge in less than $350$ hours.

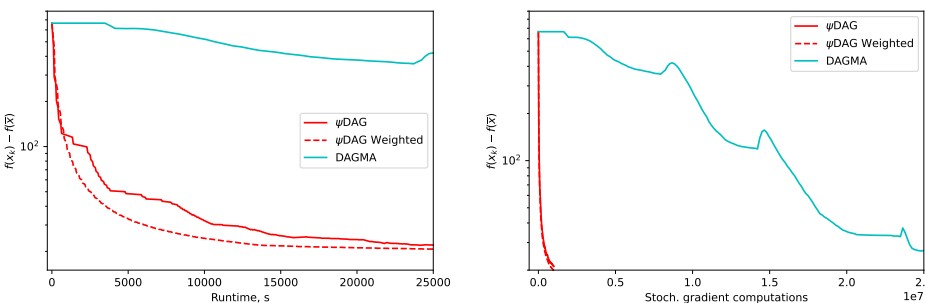

Figure 21: Comparison of $\psi$DAG, $\psi$DAG weighted and DAGMA for ER2 graph with $d = 3000$ nodes and Gaussian noise.

# E WEIGHTED PROJECTION

Inspired by the importance sampling, we considered adjustment of the projection method by weights. Specifically, we considered the elements of the $\mathbf{W}$ to be weighted element-wisely by the second directional derivatives of the objective function, $\mathbf{L}[i][j] \overset{def}{=} \left( \frac{d}{d\mathbf{W}[i][j]} \right)^2 \mathbb{E}_{X \sim \mathcal{D}} \left[ l(\mathbf{W}; X) \right]$. As we do not have access to the whole distribution $\mathcal{D}$, we approximate it by the mean of already seen samples,

$$\mathbf{L}_k[i][j] \overset{def}{=} \left( \frac{d}{d\mathbf{W}[i][j]} \right)^2 \frac{1}{k} \sum_{k=0}^{k-1} l\left(\mathbf{W}; X_k\right) = \frac{1}{k} \sum_{t=0}^{k-1} \left( X_k[j] \right)^2. \tag{19}$$

Weights (19) are identical for whole columns; hence, they impose storing only one vector. Updating them requires a few element-wise vector operations.

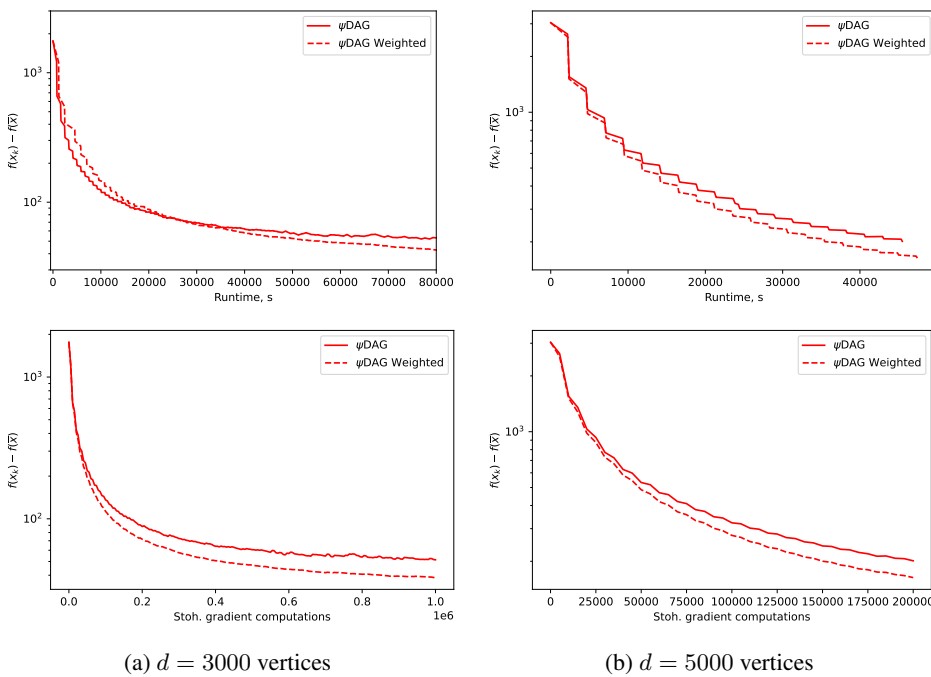

(a) $d = 3000$ vertices            (b) $d = 5000$ vertices

Figure 22: $\psi$DAG method with weighted projection for graph types ER4 and Gaussian noise.

Figures 21 and 22 show that this weighting can lead to an improved convergence (slightly faster convergence to a slightly lower functional value) without imposing any extra gradient computation. However, we noticed that the improvement over runtime is not consistent across different experiments; hence, for simplicity, we deferred this to the appendix.

