# OpenReview forum: "$\psi$DAG: Projected Stochastic Approximation Iteration for Linear DAG Structure Learning"
_ICLR.cc/2026/Conference — Submitted to ICLR 2026_

### Official Review · Reviewer_u9Q7 · 2025-10-27

**Soundness:** 3
**Presentation:** 2
**Contribution:** 2
**Rating:** 4
**Confidence:** 3

**Summary:**

The paper proposes ψDAG, a framework for Directed Acyclic Graph (DAG) structure learning based on Stochastic Approximation (SA) principles combined with projected stochastic gradient methods. The authors reformulate the DAG learning problem as a stochastic optimization task and introduce projection-based steps to enforce acyclicity efficiently. They claim scalability to large graphs (up to 10,000 nodes) and show empirical results comparing ψDAG with NOTEARS, GOLEM, NOCURL, and DAGMA.

**Strengths:**

1- The theoretical presentation is mostly sound; I did not find explicit errors in the mathematical derivations or proofs.


2- The idea of combining stochastic approximation (SA) with projection-based DAG constraints is straightforward and could be computationally appealing.


3- The authors provide comparisons with several standard baselines, including NOTEARS, GOLEM, and DAGMA.


4- The proposed algorithm is simple and might be useful in certain large-scale linear SEM settings.

**Weaknesses:**

1- The paper presents two main ideas:
(1) a new formulation of a stochastic loss function (Eq. 9) and an equivalent version based on the adjacency matrix (Eq. 10), and
(2) a strategy to reduce runtime by first learning the graph skeleton through a stochastic optimization procedure, then estimating a variable ordering via a heuristic projection function, and finally constructing the best DAG consistent with that ordering.

Regarding the first idea, the method largely reuses standard stochastic approximation updates and applies them to DAG learning with only minor modifications compared to existing approaches such as NOTEARS. For the second idea, the proposed projection mechanism is purely heuristic and lacks both theoretical justification and novelty relative to prior constrained optimization methods. Moreover, the paper overlooks several related approaches, such as BOSS, which first determines a variable ordering and then performs score-based structure learning with a BIC score and greedy search, as well as various hybrid methods that use constraint-based algorithms to obtain an initial DAG or skeleton for subsequent optimization. These omissions weaken the claimed novelty and contextual positioning of the work.

2- The reported numerical results are unconvincing. The claimed scalability (10,000 nodes) is not backed by verifiable or reproducible evidence. The GitHub link to the implementation (https://anonymous.4open.science/r/psiDAG-8F42) appears to be non-functional, which undermines reproducibility and transparency.

3- The Sachs protein network results are particularly unconvincing: ψDAG reports a Structural Hamming Distance (SHD) of 14, which is not competitive. Simple score-based or constraint-based methods, such as PC or Hill-Climbing (HC), can achieve lower SHD on this dataset.

4- The paper suffers from weak writing quality, with numerous typographical errors and noticeable inconsistencies and discontinuities between paragraphs.

5- The method is limited to linear Bayesian (SEM) models only. This is a strong limitation, especially since many recent DAG-learning frameworks (e.g., nonlinear NOTEARS, DAG-GNN) address nonlinear dependencies.

**Questions:**

1- Can the proposed method be extended to handle categorical variables in the model, or is it limited to continuous (linear Gaussian) settings?

2- Is there any theoretical justification or proof showing that the proposed projection function can reliably recover the correct variable ordering?

3- Could the authors evaluate the method on additional benchmark datasets, such as ALARM, Link, and Munin, available at https://www.bnlearn.com/bnrepository/?

---

> ### Author Response · Authors · 2025-11-21
> **Rebuttal by Authors Part 1**
>
> *We thank the reviewer for recognizing the **theoretical soundness**, **computational efficiency**, and **practical scalability** of $\Psi$DAG in large-scale linear SEM settings. Your insights are invaluable, and we believe that addressing them will significantly strengthen our work.*
>
> > W1: Regarding the first idea, the method largely reuses standard stochastic approximation updates and applies them to DAG learning with only minor modifications compared to existing approaches such as NOTEARS. For the second idea, the proposed projection mechanism is purely heuristic and lacks both theoretical justification and novelty relative to prior constrained optimization methods. Moreover, the paper overlooks several related approaches, such as BOSS, which first determines a variable ordering and then performs score-based structure learning with a BIC score and greedy search, as well as various hybrid methods that use constraint-based algorithms to obtain an initial DAG or skeleton for subsequent optimization. These omissions weaken the claimed novelty and contextual positioning of the work.
>
>
> We would like to clarify our perspective on the problem to understand whether you agree or not. The original problem of DAG-learning is proved to be **NP-hard**, even for the linear case. The DAG constraint is essentially a discrete optimization constraint combined with a continuous optimization constraint. This makes all methods, which aim to solve the original problem, **heuristics or meta-heuristics**, as it is impossible to find the global solution fast enough. The best we can aim for from an optimization perspective is local minima or stationary points without any guarantee to converge to the global solution. Note that all state-of-the-art methods, like NOTEARS, GOLEM, NOCURL, and DAGMA, can be called meta-heuristics as there is no guarantee that they converge to the global minimum.
>
> In our paper, we aim for a rather practical goal, to **find a feasible DAG close to the solution for big or huge-sized problems in a reasonable time**. While theoretical global guarantees are out of reach, our experiments demonstrate that $\Psi$DAG achieves this goal, providing robust and efficient structure recovery on graphs up to 10,000 nodes.
>
> Regarding the BOSS paper, the authors perform a permutation-based search using the BIC score. The algorithm searches over variable orderings and identifies a representative DAG  of the underlying Markov equivalence class. In contrast,  our proposed method performs continuous stochastic optimization over the weighted adjacency matrix and directly enforces acyclity via projection rather than permutation-based equivalence. We will include the paper in the related work section. We appreciate the reviewer’s suggestion and would be glad to include any additional hybrid approaches they consider relevant to ensure the related-work discussion is as complete and balanced as possible.
>
> >W2: The reported numerical results are unconvincing. The claimed scalability (10,000 nodes) is not backed by verifiable or reproducible evidence. The GitHub link to the implementation (https://anonymous.4open.science/r/psiDAG-8F42) appears to be non-functional, which undermines reproducibility and transparency.
>
> We thank the reviewer for bringing this to our attention. We are sorry that the link was not working. We believe we had some issues with anonymous.4open. We’ve fixed them now, and the link should be fully functional (https://anonymous.4open.science/r/psiDAG-8F42). All code, configuration files, and scripts used in our experiments are publicly available there to ensure full reproducibility and transparency.
> If there are specific details or additional materials the reviewer would like us to include to facilitate verification, we would be happy to provide them.
>
> >W3: The Sachs protein network results are particularly unconvincing: $\Psi$DAG reports a Structural Hamming Distance (SHD) of 14, which is not competitive. Simple score-based or constraint-based methods, such as PC or Hill-Climbing (HC), can achieve lower SHD on this dataset.
>
> The Sachs dataset was included primarily to validate $\Psi$DAG on a widely used real-world benchmark and to enable comparison with other continuous optimization based methods such as NOTEARS, GOLEM, and NOCURL. Since this dataset is relatively small, our focus was not on optimizing for the best possible SHD but on demonstrating that $\Psi$DAG performs comparably to similar differentiable methods while maintaining scalability to much larger graphs. We agree that constraint-based or score-based methods, such as PC or Hill-Climbing, can achieve lower SHD on this small-scale benchmark.

---

> ### Author Response · Authors · 2025-11-21
> **Rebuttal by Authors Part 2**
>
> > W4: The paper suffers from weak writing quality, with numerous typographical errors and noticeable inconsistencies and discontinuities between paragraphs.
>
> We thank the reviewer for the feedback. Please kindly point out the specific typos or discontinuities you identified, and we will correct them accordingly. Without the specific examples, it is hard for us to improve the paper.
>
> >W5: The method is limited to linear Bayesian (SEM) models only. This is a strong limitation, especially since many recent DAG-learning frameworks (e.g., nonlinear NOTEARS, DAG-GNN) address nonlinear dependencies.
>
> $\Psi$DAG focuses on linear SEMs, as our goal is to develop a **scalable and efficient optimization framework for large-scale DAG learning**. Even in the linear setting, learning accurate structures on graphs with thousands of nodes remains computationally challenging and has not been addressed by prior existing methods. For example, for $d=10000$ nodes, we have a total around $10^{10}$ parameters. Introducing nonlinear dependencies would further increase both model complexity and computational cost.
>
> Notably, nonlinear variants of NOTEARS and DAG-GNN, while valuable, are typically limited to much smaller graphs than their linear counterparts. Similar to the progression of the NOTEARS line of work, where the initial focus was on the linear, continuous formulation before extending to nonlinear models, we first aim to establish the scalability and stability of the proposed framework in the linear case. Extending $\Psi$DAG to nonlinear settings is a natural next step, which we plan to explore in future work as discussed in the Limitations paragraph.
>
> >Q1: Can the proposed method be extended to handle categorical variables in the model, or is it limited to continuous (linear Gaussian) settings?
>
> We thank the reviewer for this question. $\Psi$DAG currently operates under the linear SEM framework for continuous data, consistent with prior differentiable DAG-learning methods such as NOTEARS, GOLEM, NOCURL, and DAGMA, which also assume continuous variables. Extending $\Psi$DAG to categorical or mixed-type data is a natural and promising direction for future work.
>
> >Q2: Is there any theoretical justification or proof showing that the proposed projection function can reliably recover the correct variable ordering?
>
> Recovering the exact variable ordering from observational data is an **NP-hard problem**, and as such, no existing method, including ours or prior continuous approaches such as NOTEARS, GOLEM,  DAGMA, and NOCURL, can guarantee recovery of the true order in polynomial time. The proposed projection function is designed as a **practical heuristic** that efficiently enforces acyclicity and provides a plausible ordering consistent with the current weighted adjacency matrix.
>
> >Q3: Could the authors evaluate the method on additional benchmark datasets, such as ALARM, Link, and Munin, available at https://www.bnlearn.com/bnrepository/?
>
> The ALARM, Link, and Munin datasets from the BNLearn repository are well-known categorical benchmarks. Since $\Psi$DAG is currently formulated for continuous data within the linear SEM framework, these datasets are not directly compatible with our present implementation.

---

### Official Review · Reviewer_tyJK · 2025-10-28

**Soundness:** 3
**Presentation:** 2
**Contribution:** 3
**Rating:** 4
**Confidence:** 4

**Summary:**

This paper proposes ψDAG, a novel framework for learning Directed Acyclic Graph (DAG) structures based on Stochastic Approximation (SA) integrated with SGD-based optimization. The key contributions include: (1) reformulating the discrete DAG learning problem as a stochastic optimization problem (Eq. 9), (2) introducing a three-stage algorithmic framework alternating between unconstrained optimization, projection onto DAG space, and constrained optimization, and (3) demonstrating scalability to graphs with up to 10,000 nodes.

**Strengths:**

1. The method successfully handles graphs with up to 10,000 nodes, significantly outperforming baselines (GOLEM/NOCURL fail beyond 3,000 nodes, DAGMA beyond 5,000).
2. The paper includes extensive experiments across multiple dimensions: graph types (ER, SF), densities (k=2,4,6), sizes (d=10 to 10,000), and noise distributions (Gaussian, Exponential, Gumbel), with both equal and non-equal variance settings.
3. The proposed projection method (Algorithm 3) has O(d²) complexity and avoids expensive matrix exponentials or log-determinant computations required by prior methods.

**Weaknesses:**

1. Theorem 8 claims convergence to a local minimum, but the proof is informal and hand-wavy. The two-case analysis doesn't rigorously establish convergence, and there's no guarantee the method won't cycle between subspaces indefinitely.
2. Section 3.1 states "which implies that the minimizer of (9) recovers the true DAG" but provides no rigorous proof. The algebraic manipulation ||x - W^⊤x|| = ||(I-W)(I-W*^⊤)^{-1}N_i|| doesn't obviously imply W=W* is the unique minimizer.
3. Unlike recent DAG learning theory (e.g., Gao et al. 2022b, Deng et al. 2023b), this paper provides no sample complexity bounds or finite-sample convergence rates.
4. Algorithm 3's greedy heuristic has no theoretical analysis. Why should minimizing row/column norms find a good topological ordering?
5. DAGMA "fails to converge" in numerous settings (protein dataset, r>15, d≥5000 for ER2). This is highly unusual given DAGMA's reported robustness in the original paper. Have implementations been verified against original codebases?
6. The paper uses a non-standard convergence criterion (f(x_k) - f(x*) ≤ 0.1·f(x*)) which requires knowing f(x*). How is this computed? Different methods may have different sensitivities to this threshold.
7. Only one small real dataset (d=11, n=853) is tested. For a method claiming scalability, evaluation on larger real networks is essential.
8. Why alternate between unconstrained optimization, projection, and constrained optimization? Why not just project once? The paper provides no theoretical or empirical justification for this design choice.
9. How are τ₁ and τ₂ chosen? How many outer iterations K are needed? What initialization is used?
10. Lemmas 2, 5, 6 are basic set theory facts that add little value. The claim that D is a conic set (Lemma 2) is trivial since scaling edge weights doesn't create cycles.

**Questions:**

Please see Weaknesses.

---

> ### Author Response · Authors · 2025-11-29
> **Rebuttal by Authors Part 1**
>
> *We thank the reviewer for the positive assessment of $\Psi$DAG and for highlighting its **scalability**, **breadth of experimental evaluation**, and the **efficiency of our projection method**. We appreciate the constructive feedback and are glad that the strengths of our approach were recognized.*
>
> >W1: Theorem 8 claims convergence to a local minimum, but the proof is informal and hand-wavy. The two-case analysis doesn't rigorously establish convergence, and there's no guarantee the method won't cycle between subspaces indefinitely.
>
> We appreciate the reviewer’s concern, and we partially agree with the statement. In the theorem’s proof, we consider two cases: (1) the minimum found within a fixed subspace is already a local minimum for the full problem, or (2) in rare cases, this minimum lies in another subspace that contains a better minimum. To prevent the method from cycling between such subspaces, we note in lines 892–893 that "*To fully guarantee convergence, one can record the visited subspaces and forbid projecting onto them again.*" This mechanism eliminates the possibility of cycles. We did not implement this safeguard in practice, as the method consistently behaved well without it
>
> >W2: Section 3.1 states "which implies that the minimizer of (9) recovers the true DAG," but provides no rigorous proof. The algebraic manipulation ||x - W^⊤x|| = ||(I-W)(I-W*^⊤)^{-1}N_i|| doesn't obviously imply W=W* is the unique minimizer.
>
> We thank the reviewer for this precise and helpful comment. First, there is a misprint in the formula; it should be $\| x - W^{T} x\| = \|(I-W^{T}) (I - W_*^{T})^{-1}N_i\|$. Second, the text in Section 3.1 is intended as an intuitive explanation rather than a formal proof.
> The algebraic relation $\| x - W^{T} x\| = \|(I-W^{T}) (I - W_*^{T})^{-1}N_i\|$ illustrates that the expected loss is minimized at $W = W_*$ since in that case the loss reduces to $\|N_i\|$, analogous to the original DAG problem. However, we agree that this argument alone does not rigorously establish that (9) recovers the true minimizer. We will therefore add a corresponding lemma in the Appendix that formally proves that $W^{\ast}$ is indeed the global minimizer of (9).
>
> >W3: Unlike recent DAG learning theory (e.g., Gao et al. 2022b, Deng et al. 2023b), this paper provides no sample complexity bounds or finite-sample convergence rates.
>
> [1] and [2] study DAG learning in highly structured, parametric settings (e.g., equal-variance Gaussian SEMs), where minimax sample-complexity analysis is tractable. PSIDAG addresses a fundamentally different and substantially more general problem: identifying the causal DAG under nonparametric additive-noise models with scalable algorithms. In this general setting, sharp sample-complexity or finite-sample guarantees are not known for any existing method and remain outside the scope of current theory. Our contribution is therefore provable correctness and consistency under broad assumptions, rather than rate-optimality in restricted parametric regimes.
>
> >W4: Algorithm 3's greedy heuristic has no theoretical analysis. Why should minimizing row/column norms find a good topological ordering?
>
> We thank the reviewer for this insightful comment. Algorithm 3 is indeed a meta-heuristic and is not supported by a formal theoretical analysis. The idea is to approximate a valid topological ordering by iteratively removing vertices with small incoming or outgoing weighted norms. In a DAG, sources (nodes with weak or few incoming edges) and sinks (nodes with weak or few outgoing edges) naturally correspond to small column and row norms of the weighted adjacency matrix. Minimizing these norms, therefore, provides a simple way to identify nodes that are likely to appear early or late in a topological ordering.
>
> While we do not claim formal optimality, this heuristic reflects a structural property of acyclic graphs: nodes with sparse or weak connections tend to be extremal in the causal order. Empirically, the method yields consistent and stable orderings across datasets, as shown by the convergence behavior of $\Psi$DAG (Figures 2–5). Its simplicity also enables an $O(d^2)$ implementation, which is important for scalability.
>
> Finally, our framework does not depend on this specific heuristic. Any alternative ordering procedure with a stronger theoretical justification can be modularly substituted without modifying the rest of the algorithm
>
> [1] Ming Gao, Wai Ming Tai, and Bryon Aragam.Optimal estimation of gaussian dag models, 2022b.
>
> [2] Chang Deng, Kevin Bello, Pradeep Ravikumar, and Bryon Aragam.Global optimality in bivariate gradient-based DAG learning. Advances in Neural Information Processing Systems, 36:1792917968,2023b.

---

> ### Author Response · Authors · 2025-11-30
> **Rebuttal by Authors Part 2**
>
> >W5: DAGMA "fails to converge" in numerous settings (protein dataset, r>15, d≥5000 for ER2). This is highly unusual given DAGMA's reported robustness in the original paper. Have implementations been verified against original codebases?
>
> Yes, we used the official DAGMA implementation from the original codebase without modification. We also verified the installation and hyperparameter settings against the authors’ recommended configurations.
>
> >W6: The paper uses a non-standard convergence criterion (f(x_k) - f(x*) ≤ 0.1·f(x*)) which requires knowing f(x*). How is this computed? Different methods may have different sensitivities to this threshold.
>
> First, we emphasize that this convergence criterion is used only for performance visualization, and $f(x^{\ast})$ is a fixed reference value shared across all algorithms for a given problem. Importantly, $f(x^{\ast})$ is **not** the unknown global optimum. Instead, it is a **reference loss evaluated at the ground-truth DAG** (or its empirical approximation), which provides a well-defined baseline for synthetic experiments. In our implementation, we compute $ f(x^*) = \frac{1}{2n}\|X - X B\|_F^2$, where $B$ is the ground-truth adjacency matrix. Thus, $f(x^{\ast})$ is evaluated, not assumed, and is identical across all methods.
>
> The threshold $f(x_k) - f(x^{\ast}) \leq 0.1 \cdot f(x^{\ast})$ is used to report runtime, i.e., the time each method needs to reach a loss within 10\% of the ground-truth baseline. It has no impact on model training or on evaluation metrics such as SHD, precision, or recall.
>
> Finally, using the loss at the ground-truth parameters as a reference is a common benchmarking practice in structure-learning and stochastic-optimization experiments on synthetic data, as it enables fair and consistent comparison of convergence speeds across methods with inherently different objective scales.
>
> >W7: Only one small real dataset (d=11, n=853) is tested. For a method claiming scalability, evaluation on larger real networks is essential.
>
> The Sachs protein network (d = 11, n = 853) is indeed the standard real benchmark in continuous-DAG literature (e.g., NOTEARS, GOLEM, NOCURL) because it is one of the few real datasets with a biologically validated ground-truth causal structure. To ensure a fair and directly comparable evaluation, we used the same dataset and protocol as prior methods. Larger real datasets typically lack an agreed-upon ground-truth DAG, making quantitative metrics such as SHD or TPR impossible to compute. For this reason, scalability was evaluated on large synthetic networks (up to 10,000 nodes), where the true structure is known, and scalability can be objectively measured.
>
> >W8: Why alternate between unconstrained optimization, projection, and constrained optimization? Why not just project once? The paper provides no theoretical or empirical justification for this design choice.
>
> First, let us clarify the design choices behind the framework:
> 1. **Unconstrained optimization:** This part is essential because it allows us to directly minimize the loss under a relaxation of the DAG constraints. However, used alone, it is not sufficient, since the resulting solution need not correspond to a DAG.
>
> 2. **Projection step:** A projection is required (at least once) to enforce the DAG constraints. However, if we project only once and then fix the resulting topological ordering, we are restricted to finding a minimizer within that single ordering and cannot explore potentially better orderings.
>
> 3. **Optimization within a fixed topological ordering:** Optimizing within a given ordering fully leverages the graph structure and can efficiently find a minimizer in that restricted space. However, such a minimizer is only local with respect to the space of all DAGs, and without the other components, the method is limited.
>
> Overall, the first two components can be seen as a mechanism for escaping local minima associated with a particular ordering. While this is not a strong theory in itself, the alternation between unconstrained optimization and projection steps is a well-known strategy in constrained optimization, closely related to projected gradient descent methods.

---

> > ### Author Response · Authors · 2025-11-30
> > **Rebuttal by Authors Part 3**
> >
> > >W9: How are τ₁ and τ₂ chosen? How many outer iterations K are needed? What initialization is used?
> >
> > $\tau_1$ and $\tau_2$ are hyperparameters of the algorithm. The number of outer iterations $K$ is also a hyperparameter that depends on the available computational budget. A larger $K$ yields a more accurate solution, while a smaller $K$ reduces runtime. In practice, we use the same values as in our experimental setup. Figure 11 illustrates how the runtime scales with $K$ and shows that ψDAG reaches the target accuracy well before the maximum number of outer iterations, demonstrating that moderate values of
> > $K$ is sufficient.
> >
> > For initialization, we use $W = 0$, which is standard in continuous
> > DAG-learning methods.
> >
> > >W10: Lemmas 2, 5, 6 are basic set theory facts that add little value. The claim that D is a conic set (Lemma 2) is trivial since scaling edge weights doesn't create cycles.
> >
> > We agree that Lemmas 2, 5, and 6 are known and summarize basic set properties, and we included them mainly for completeness and to make later proofs self-contained. These lemmas formally establish that the DAG space $\mathcal{D}$ can be decomposed into linear subspaces associated with topological orderings, an essential step for Theorems 7 and 8.

---

### Official Review · Reviewer_Hg8k · 2025-11-01

**Soundness:** 4
**Presentation:** 3
**Contribution:** 3
**Rating:** 6
**Confidence:** 4

**Summary:**

This paper focuses on the problem of learning graphical structures (Directed Acyclic Graphs; DAGs) from data, specifically targeting the typical linear model structure learning frameworks used in existing methods like NOTEARS (Zheng et al., NeurIPS 2018), GOLEM (Ng et al., NeurIPS 2020), and NOCURL (Yu et al., ICML 2021). The paper proposes a new algorithm called 𝜑-DAG. The key idea is a three-stage optimization process: unconstrained optimization → projection onto the DAG space → optimization that preserves the vertex order. Stochastic gradient methods are applied in both the first and third stages. This approach reduces the search space from all possible DAGs, an exponentially large space, to the space of topological orderings, enabling a more efficient algorithm suited for large-scale problems. Empirical results show that 𝜑-DAG outperforms existing methods like NOTEARS, GOLEM, and NOCURL in comparative experiments.

**Strengths:**

- This paper presents a very interesting and robust algorithm that tackles one of the key challenges in DAG structure learning, i.e. how to satisfy the strict DAG constraint, which has been a major difficulty for existing optimization methods. The proposed approach decomposes the problem into largely independent subproblems: optimization → projection onto the constraint-satisfying solution space → (weakly-)constrained optimization, forming a three-stage framework.
- In the final step (step 3) of this three-stage process, optimization must be performed while preserving the topological order of vertices determined in step 2. To handle this, the paper introduces a valid method based on computing the transitive closures and applying masking to enforce the order constraints within optimization.

- Previous methods that needed to handle strict constraints over an exponentially large DAG search space, but the proposed method with Algorithm 3 used in step 2 can now reduce the problem to enforcing strict constraints over node orderings in step 3. This effectively narrow down the search from the exponentially combinatorial space of DAGs to the permutation space of node orderings, which is smaller, and leads to a more efficient and logically grounded solution in the proposed method.

- Experimental results demonstrate empirical superiority in both accuracy and computational efficiency compared to representative existing methods, including NOTEARS (Zheng et al., NeurIPS 2018), GOLEM (Ng et al., NeurIPS 2020), and NOCURL (Yu et al., ICML 2021).
- Each of the points is thoroughly explained in the appendix, which is more than great.

**Weaknesses:**

- Since the idea of using stochastic gradient descent and the idea of using projection methods seem largely independent, an ablation study analyzing the contribution of each would make the work more informative. For example, in non-convex hard-constraint optimization problems like those with L0-norm penalties, projected gradient methods are a traditional approach. However, it’s well known that even simple gradient descent combined with projection often faces challenges in terms of convergence guarantees and optimality. These issues typically require additional techniques or relaxations, and simply replacing the gradient method with a stochastic version likely doesn’t resolve them on its own.

- The SI provides detailed explanations, but a clearer discussion in the main text about how existing methods handle hard DAG constraints and how the proposed method takes a different approach would help readers better understand the contributions of this work.

- The rationale for using stochastic gradient methods from the perspective of Stochastic Approximation (SA) vs. Sample Average Approximation (SAA) is valid as described, but a bit misleading. In practice, the difference between the sample average and the expectation is often handled with some form of regularization in SAA. So, while adopting SA may offer benefits in terms of computational efficiency or convergence stability, the current explanation suggesting it directly improves approximation accuracy may be a bit confusing. That said, recent work has shown that stochastic gradient methods can offer implicit regularization in complex optimization landscapes, so this could be useful to clarify the benefit with a more careful explanation.

- It seems the formulation reuses a standard setup, but since the objective function implicitly becomes quadratic, it would be helpful to include a brief explanation. When the noise term ( N ) is Gaussian, a quadratic objective is appropriate. However, for cases like exponential or Gumbel noise, as tested in the experiments, it’s not immediately clear whether the quadratic objective is still valid. One possible reason for the proposed method’s stability might be that, while the DAG constraint is complex, the error term’s quadratic form provides favorable properties, and this could be indirectly contributing to its effectiveness.

**Questions:**

I'm not a researcher in this specific area, so I'd like to ask a few clarification questions:

- From a general optimization design perspective, is the main takeaway that, in the case of DAG constraints, methods that explicitly account for graph structure are more effective, meaning that standard approaches like Projected Gradient Methods or Proximal Gradient Methods with convex relaxations are not sufficient?

- On p.15 of the appendix, are the objective functions in the existing methods optimized using techniques other than stochastic gradient methods? Since the objective function and the optimization strategy are conceptually separate, it seems that one could, in principle, apply stochastic gradient methods to equations (11), (12), and (13) by handling constraints via Lagrangian multipliers and using proximal methods for the L1 terms. Was this tested? Or is there some technical barrier that makes introducing stochastic gradient methods into this problem particularly challenging? A clearer explanation of this point would be appreciated.

- Both the existing formulations and the proposed method use a quadratic loss term, but is there no assumption of Gaussian noise? As shown on p.7, the noise ( N ) is tested not only with Gaussian noise, but also with Exponential and Gumbel noise. In those cases, wouldn’t a linear loss be more appropriate for Exponential noise, and a logistic loss for Gumbel noise? Wouldn't this part affect the entire paper?

- In lines 290–293 on page 6, it says, “if we know the true topological ordering ord(G∗), then we can recover the true DAG W∗ with high accuracy.” However, in practice, we don’t actually know the true topological ordering, and we can't guarantee that the node ordering obtained in Step 2 is the true one. So, should we understand this not as a theoretical guarantee of finding the exact solution, but rather as a claim that the search space has been reduced from the combinatorial DAG space to the permutation space of node orderings?

- Since the true topological ordering generally can't be identified, that means even when using the proposed method, if the ordering obtained in Step 2 isn't the true one, the solution won't converge to the correct one, as we can see in Figure 2, right? I’d appreciate it if you could provide some clarification, as the takeaway in Section 4.1 wasn’t entirely clear to me.

---

> ### Author Response · Authors · 2025-11-21
> **Rebuttal by Authors Part 1**
>
> *We sincerely thank the reviewer for their detailed feedback. We appreciate the positive evaluation of our algorithm’s **soundness**, **clarity**, and **contribution**, as well as the recognition of $\Psi$DAG’s effectiveness in addressing the DAG constraint and **improving scalability**. Your insights are invaluable, and we believe that addressing them will significantly strengthen our work.*
>
> >W1: Since the idea of using stochastic gradient descent and the idea of using projection methods seem largely independent, an ablation study analyzing the contribution of each would make the work more informative. For example, in non-convex hard-constraint optimization problems like those with L0-norm penalties, projected gradient methods are a traditional approach. However, it’s well known that even simple gradient descent combined with projection often faces challenges in terms of convergence guarantees and optimality. These issues typically require additional techniques or relaxations, and simply replacing the gradient method with a stochastic version likely doesn’t resolve them on its own.
>
> We thank the reviewer for the insightful comment and agree that understanding the relative contributions of stochastic optimization and projection is valuable. However, we chose not to include such an ablation study due to its significant computational overhead and our focus on the main results. In particular, full-batch GD with projection would be substantially slower and more computationally expensive, even just in terms of optimizing the quadratic loss.
>
> Below, we clarify why, although stochastic approximation and projection are conceptually independent, their **combination is essential** for the $\Psi$DAG framework.
> **Projection enforces feasibility, and SGD provides scalability and robustness**. The projection step guarantees that every iterate remains within the acyclic domain, completely removing the need for penalty-based acyclicity functions used in NOTEARS or GOLEM. From the practical point of view, it is quite common that GD faces more issues in non-convex optimization compared to SGD. SGD is more explorative compared to GD and can better escape stationary points and even find a better local minimum for the non-convex problem.
>
> >W2: The SI provides detailed explanations, but a clearer discussion in the main text about how existing methods handle hard DAG constraints and how the proposed method takes a different approach would help readers better understand the contributions of this work.
>
> We thank the reviewer for the helpful suggestion, and we will revise the manuscript accordingly.
>
> >W3: The rationale for using stochastic gradient methods from the perspective of Stochastic Approximation (SA) vs. Sample Average Approximation (SAA) is valid as described, but a bit misleading. In practice, the difference between the sample average and the expectation is often handled with some form of regularization in SAA. So, while adopting SA may offer benefits in terms of computational efficiency or convergence stability, the current explanation suggesting it directly improves approximation accuracy may be a bit confusing. That said, recent work has shown that stochastic gradient methods can offer implicit regularization in complex optimization landscapes, so this could be useful to clarify the benefit with a more careful explanation.
>
> We thank the reviewer for this insightful comment. As we mentioned before, the SGD-based optimization method is crucial for the performance of $\Psi$DAG. Regarding the SA versus SAA discussion, our framework can, in principle, operate under both formulations. We decided to focus more on the SA part to highlight the benefits of SGD-type methods and bring the attention of the causality community to it, as most SOTA methods (NOTEARS, GOLEM, DAGMA, etc) use full batch methods over SGD based. We also agree with the reviewer’s observation that stochastic gradients provide implicit regularization in complex nonconvex landscapes, and we will incorporate this clarification into the revised manuscript.

---

> > ### Author Response · Authors · 2025-11-21
> > **Rebuttal by Authors Part 2**
> >
> > >W4: It seems the formulation reuses a standard setup, but since the objective function implicitly becomes quadratic, it would be helpful to include a brief explanation. When the noise term ( N ) is Gaussian, a quadratic objective is appropriate. However, for cases like exponential or Gumbel noise, as tested in the experiments, it’s not immediately clear whether the quadratic objective is still valid. One possible reason for the proposed method’s stability might be that, while the DAG constraint is complex, the error term’s quadratic form provides favorable properties, and this could be indirectly contributing to its effectiveness.
> >
> > We thank the reviewer for this thoughtful observation and agree that the quadratic loss corresponds exactly to the Gaussian noise assumption.
> >
> > In DAG learning, the primary objective is to recover the correct adjacency structure rather than to model the full data likelihood. Under the additive noise model $X = X W^{*} + N,  \mathbb{E}[N] = 0,\; \mathrm{Cov}(N) = \mathrm{diag}(\sigma_1^2,\dots,\sigma_d^2),$
> >
> > the true DAG $W^{*}$ minimizes the expected quadratic risk $\mathcal{R}(W) = \mathbb{E}\left[\|X - XW\|_2^2\right] $,
> >
> >  even when the noise distribution of $N$ is non-Gaussian. This holds because, for each variable $X_j$, the conditional expectation $ \mathbb{E}[X_j \mid X_{\mathrm{Pa}(j)}]$ remains a linear function of its parents under the linear SEM assumption. Consequently, the optimal regression coefficients coincide with the true structural weights $W^{*}$, regardless of the specific distribution of $N$. Hence, minimizing the quadratic loss serves as a distribution-agnostic surrogate that consistently recovers the correct DAG structure while providing a smooth and well-conditioned objective for optimization.
> >
> >
> > >Q1: From a general optimization design perspective, is the main takeaway that, in the case of DAG constraints, methods that explicitly account for graph structure are more effective, meaning that standard approaches like Projected Gradient Methods or Proximal Gradient Methods with convex relaxations are not sufficient?
> >
> > Yes, this is one of the takeaways. First, because it is quite challenging to perform a classical exact $l_2$ projection, as the projection should be onto the DAG constraint. In the projection subproblem, $\min \|W-W_k\|_2$ such that $W \in DAG$, we simplified the objective, but the constraint remains, and this constraint is precisely the main difficulty of the original problem. By incorporating the graph structure and explicit set geometry, we can perform more effective projections onto a smaller subspace of orderings.
> >
> > >Q2: On p.15 of the appendix, are the objective functions in the existing methods optimized using techniques other than stochastic gradient methods? Since the objective function and the optimization strategy are conceptually separate, it seems that one could, in principle, apply stochastic gradient methods to equations (11), (12), and (13) by handling constraints via Lagrangian multipliers and using proximal methods for the L1 terms. Was this tested? Or is there some technical barrier that makes introducing stochastic gradient methods into this problem particularly challenging? A clearer explanation of this point would be appreciated.
> >
> > We thank the reviewer for this insightful question. The objectives on p.15 are optimized by full batch GD-type methods, not SGD. While stochastic gradients could, in principle, be applied to prior formulations such as NOTEARS, GOLEM, or DAGMA (Eqs.(11)-(13)), doing so directly is computationally impractical. The main difficulty lies in the acyclicity function $h(w)$. Computing its gradient at each iteration is more expensive than evaluating the full batch itself, which has led all existing methods to rely on full-batch optimization. We believe that SGD-type approaches could, in principle, be adapted to formulations (11), (12), and (13), but this would require a more efficient treatment of the acyclicity constraint, an open direction that, to our knowledge, has not yet been addressed in the literature.
> >
> > In contrast, our method treats DAG learning as a stochastic approximation problem with sample-wise separable loss and enforces acyclicity through an efficient projection step (Algorithm 3), allowing for the stable and scalable use of stochastic gradients.

---

> > > ### Author Response · Authors · 2025-11-21
> > > **Rebuttal by Authors Part 3**
> > >
> > > >Q3: Both the existing formulations and the proposed method use a quadratic loss term, but is there no assumption of Gaussian noise? As shown on p.7, the noise ( N ) is tested not only with Gaussian noise, but also with Exponential and Gumbel noise. In those cases, wouldn’t a linear loss be more appropriate for Exponential noise, and a logistic loss for Gumbel noise? Wouldn't this part affect the entire paper?
> > >
> > > The quadratic loss is a standard, distribution-agnostic choice for DAG structure recovery; the non-Gaussian experiments serve to demonstrate robustness rather than to imply a misspecified likelihood assumption.
> > > We thank the reviewer for this thoughtful observation. For the Gaussian noise, the quadratic loss corresponds to the maximum likelihood problem. For other types, the quadratic loss in Eq. (2) is used as a general least-squares score for structure recovery, not as an assumption of Gaussian noise. In linear SEM-based DAG learning, the acyclicity constraint, not the specific noise distribution, governs the recovery of edge directions. Using a quadratic loss is therefore standard across prior continuous methods (NOTEARS, GOLEM, NOCURL, and  DAGMA) even when non-Gaussian or heteroscedastic noises are present.
> > > The experiments with Exponential and Gumbel noises are designed to test the robustness of structure recovery under distributional misspecification. $\Psi$DAG maintains accurate edge recovery and convergence in these regimes, indicating that the optimization framework is insensitive to moderate deviations from Gaussianity.
> > >
> > >
> > > >Q4: In lines 290–293 on page 6, it says, “if we know the true topological ordering ord(G∗), then we can recover the true DAG W∗ with high accuracy.” However, in practice, we don’t actually know the true topological ordering, and we can't guarantee that the node ordering obtained in Step 2 is the true one. So, should we understand this not as a theoretical guarantee of finding the exact solution, but rather as a claim that the search space has been reduced from the combinatorial DAG space to the permutation space of node orderings?
> > >
> > > Yes, the true ordering is unknown, and we can’t guarantee that we will reach it in any reasonable time (it may take d! projections and some additional adaptations to not project onto the same ordering twice). These lines were meant to show that the problem of finding a true DAG is almost equivalent to finding a true ordering. Which, as you said, means that the search space has been reduced from the combinatorial DAG space to the permutation space of node orderings.
> > >
> > > >Q5: Since the true topological ordering generally can't be identified, that means even when using the proposed method, if the ordering obtained in Step 2 isn't the true one, the solution won't converge to the correct one, as we can see in Figure 2, right? I’d appreciate it if you could provide some clarification, as the takeaway in Section 4.1 wasn’t entirely clear to me.
> > >
> > > Yes, the DAG constraint is highly non-convex and may exhibit an exponential number of local minima. Moreover, it has been shown that the problem is NP-hard, so we cannot guarantee convergence to a global minimum in polynomial time. Therefore, the goal of $\Psi$DAG is to find a good feasible DAG efficiently and in a manner that scales to large graphs.

---

### Author Response · Authors · 2025-12-04
**Updated Manuscript**

We are grateful for the reviewers’ thoughtful comments and suggestions. We have incorporated Lemma 7 into the appendix to strengthen the theoretical discussion, and we have updated the related work section to reflect the reviewer’s suggestions.

---

### Meta-Review · Area_Chair_ewjy · 2026-01-06

**Summary:**

The work proposes a novel algorithm for causal discovery using a 3-stage optimization process, namely, 1. unconstrained optimization 2. projection onto the DAG space and 3. optimization preserving the ordering of the vertices and thereby preserving acyclicity. The experiments are conducted on a synthetic dataset as well as the real-world causal protein signaling network. Another important aspect is the scalability since the synthetic data set experiments are conducted even for the graph size of 10,000.

The paper received 3 initital reviews, with 1 reviewer being positive and the other 2 reviewers being slighlty negative about the paper. The major points of contention from the reviewers were as follows:

* Limited novel innovation. Reviewer u9Q7 points out that the proposed approach is similar to existing apporaches such as NOTEARS and the proposed projection mechanism is purely heuristic.

* Weak experimental results especially in the  Gaussian noise with non-equal variances (NV) cases which are actually the more interesting cases. Also, scalability is claimed but is confined to a synthetic setting, where getting the ordering of variables is easier. FOr real-world graphs this might not scale.

**Reviewer Concerns:**

The 1st concern of limited novelty is not answered well in my opinion. Although I do agree with the authors that the aim of their paper is more practical in nature, I do not see how it is set apart from previous methods (especially when looking at the internal mechanisms used) when looked at from an experimental angle. Theoretically no algorithm can provide any guarantee and like all previous works, the converge is only to a local minima. Also the heuristic used for projection mechanism does not guarantee correct variable ordering and I see this failing for even simple real world cases.

Also, for the experiments, limiting only to continuous variables is a drawback. I do understand that not everything can be done in a single paper, but having atleast an experiment on a categorical data set would have made the paper more strong and general in nature. Right now, I see the proposed method struggling with more complex data types and broader benchmarking.

**Reviewer Scores:**

I do not see a potential increase in the reviewer score based on the current rebuttal. I think the paper is going in the correct direction, but a lot of open questions remain. I thus recommend rejection at this point.

---

### Decision · Program_Chairs · 2026-01-26

Reject